# Global Prompt Refinement with Non-Interfering Attention Masking for One-Shot Federated Learning

**Zhuang Qi[1], Yu Pan[1], Lei Meng[1,*], Sijin Zhou[2], Han Yu[3], Xiaoxiao Li[4,5], Xiangxu Meng[1]**
[1]School of Software, Shandong University, China
[2]AIM Lab, Faculty of Engineering, Monash University, Clayton, VIC, Australia
[3]College of Computing and Data Science, Nanyang Technological University, Singapore
[4]Department of Electrical and Computer Engineering, University of British Columbia, Canada
[5]Vector Institute, Canada
z_qi@mail.sdu.edu.cn, 202100300334@mail.sdu.edu.cn, lmeng@sdu.edu.cn,
sjzhou1995@gmail.com,han.yu@ntu.edu.sg, xiaoxiao.li@ece.ubc.ca, mxx@sdu.edu.cn

## Abstract

Federated Prompt Learning (FPL) enables communication-efficient adaptation by tuning lightweight prompts on top of frozen pre-trained models. Existing FPL methods typically rely on global information, which is only available after the second training round, to facilitate collaboration among client models. Therefore, they are inherently dependent on multi-round communication to fully exhibit their strengths. Moreover, existing one-shot federated learning methods typically focus on fitting seen tasks, but lack cross-task generalization. To bridge this gap, we propose the Global Prompt Refinement with Non-Interfering Attention Masking (GPR-NIAM) method for one-shot FPL. The core idea is to design a masking mechanism that restricts excessive interaction between the original text embeddings and the learnable prompt embeddings. GPR-NIAM achieves this through the collaboration of two key modules. Firstly, the attention isolation module suppresses attention from the learnable prompt tokens to the original text tokens, and reweights the reverse attention which preserves generalization across tasks. Secondly, the cross-silo collaborative refinement module integrates decentralized visual knowledge into a unified base and calibrates the global prompt through multi-source cross-modal knowledge alignment, further mitigating the inconsistency caused by data heterogeneity. Extensive experiments conducted on ten benchmark datasets under two tasks show that GPR-NIAM outperforms eight state-of-the-art methods in both class-level and domain-level generalization.

## 1 Introduction

Federated learning (FL) enables collaborative modeling across data sources in a privacy-preserving manner, which has gained traction across multiple domains [1, 2, 3, 4, 5]. It learns global knowledge by aggregating model weights or gradients instead of moving data to a central entity [6, 7, 8, 9]. Recently, Federated Prompt Learning (FPL) has emerged to support parameter-efficient fine-tuning in this setting. It not only leverages pre-trained models (e.g., CLIP [10]) more effectively, but also reduces communication overhead between FL clients and the FL server by transmitting only learnable prompts [11, 12, 13, 14]. However, heterogeneous data distributions across clients often degrade model aggregation performance and slow down convergence [15, 16]. This is because significant differences in the update directions across local models can cause conflicts during aggregation

---

*Corresponding author

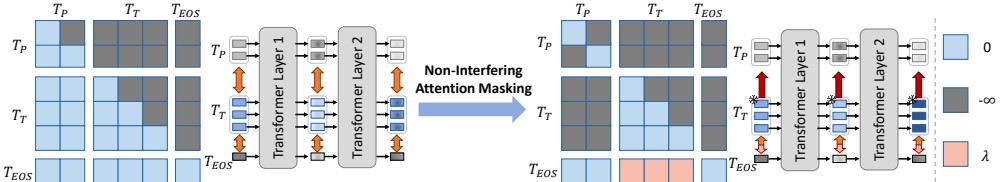

Figure 1: An overview of the proposed `GPR-NIAM` method. It supports three interaction types: full attention (0), hard masking ($-\infty$), and reweighting ($\lambda$). Hard masking blocks learnable tokens $T_P$ from affecting original text tokens $T_T$, while $\lambda$-weighted links enable partial influence from special tokens $T_{EOS}$. The upper triangular mask follows causal attention for autoregressive decoding.

[17, 18, 19, 20, 21]. As a result, existing FPL methods still require multi-round communications between clients and the server to reconcile inconsistencies arising from client-specific training.

To deal with data heterogeneity, early FPL solutions (e.g., PromptFoilo [17] and DP-FPL [22]) typically focused on incorporating global prompt information to constrain the update direction of local prompts in order to enhance the consistency and stability of federated optimization [17, 23, 22, 24]. However, since global prompt information is typically available only after the second communication round, these methods inherently rely on multi-round communications. This poses a challenge in real-world scenarios with limited communication resources. To this end, one-shot federated learning (OSFL), which requires only a single round of client-server interaction, offers a promising direction for further exploration [25, 26, 27, 28, 29]. OSFL significantly reduces the communication overhead by eliminating iterative exchanges, making it attractive for real-world deployments with strict latency or bandwidth constraints [30, 31, 32]. However, directly applying existing OSFL methods to FPL settings is not suitable, as they are typically optimized for seen tasks while neglecting the preservation of generalization to unseen tasks.

To bridge this important gap, we propose the Global Prompt Refinement with Non-Interfering Attention Masking (GPR-NIAM) method. It prevents the modification of the original token embeddings through a masking mechanism, thereby preserving transferable knowledge across tasks. As shown in Figure 1, the interaction between the learnable prompt embeddings and the original text token embeddings is unidirectional, which ensures that the latter remain unchanged. Specifically, `GPR-NIAM` consists of two main modules. Firstly, the Attention Isolation module selectively regulates the flow of prompt information through a novel masking strategy. It suppresses attention from the learnable prompt tokens to the original text tokens across all encoder layers, thereby effectively preventing alterations to the textual input token embeddings. In addition, it re-weights the reverse attention to encourage the model to extract task-relevant features more effectively from the original texts. Secondly, the cross-silo collaborative refinement (CSCR) module leverages visual representations extracted from multiple clients and utilizes multi-source visual supervision to refine the global prompt in a centralized manner. This further mitigates the negative impact of data heterogeneity.

We conduct extensive experiments on 10 datasets under two tasks: 1) base-to-base/novel generalization and 2) leave-one-domain-out generalization, and carry out performance comparisons, ablation studies, in-depth analysis and case studies. The results demonstrate that `GPR-NIAM` achieves superior overall performance by maintaining a strong balance between fitting seen tasks and generalizing to unseen ones, significantly outperforming eight competitive state-of-the-art approaches.

## 2 Related Work

### 2.1 Federated Prompt Learning

FPL aims to integrate FL with prompt learning by freezing the backbone model within the FL framework and introducing a set of learnable prompt parameters [17, 23, 22, 24, 33, 34, 35, 36]. This enables efficient personalization for on-device data while significantly reducing communication overhead, as only prompt parameters are shared between clients and the server. To mitigate the challenges posed by data heterogeneity across clients, popular FPL methods (e.g., FedOTP [24], FedPGP [23], PromptFolio [17], DP-FPL [22]) mostly adopt a shared prompt paradigm. In this design, a global prompt is constructed on the server and combined with client-specific information to

improve training stability and generalization under non-IID distributions. However, these methods are essentially designed for multi-round communications and require access to global feedback, which limits their effectiveness in one-shot FL scenarios.

## 2.2 One-Shot Federated Learning

OSFL is a communication-efficient variant of federated learning, where model training is completed within a single round of client-server interaction [25, 26, 37, 27, 28, 38, 39, 40, 41, 42]. Recent studies on OSFL mainly focus on generative and ensemble-based approaches. The former employs generative models to synthesize proxy data or representations on the server. For example, FedCVAE [41] uses a variational autoencoder to compress local knowledge. FedDISC [28] and FedDEO [38] leverage diffusion models to generate synthetic training data. The latter is dedicated to refining the fusion of model predictions from different clients. For example, FedOV [27] introduces an open-set recognition-based voting mechanism to address label skew in OSFL; IntactOFL [26] learns weighted or expert-based combinations to enhance overall performance. However, these methods are only designed for conventional tasks, aiming to fit the training objective while overlooking the preservation of cross-task generalization.

# 3 Preliminaries

Suppose there are $K$ clients denoted by $\mathcal{C} = \{C_1, C_2, \ldots, C_K\}$, and each client $C_k \in \mathcal{C}$ holds a private dataset $\mathcal{D}_k$. The goal of traditional FL is to collaboratively train a global model $f$ by minimizing the objective: $\min_f \sum_{k=1}^K p_k \cdot \mathcal{L}(f, \mathcal{D}_k)$, where $p_k = \frac{|\mathcal{D}_k|}{\sum_{j=1}^K |\mathcal{D}_j|}$ is the weight of client $k$, and $\mathcal{L}(\cdot)$ is a task-specific loss function [43, 44, 45, 46]. In FPL, each client $k$ optimizes a set of learnable prompts $\delta_k$ using its private dataset $\mathcal{D}_k$, while keeping the backbone model $f$ frozen: $\min_{\delta_k} \mathcal{L}(f; \delta_k, \mathcal{D}_k)$. To support class-level prediction, the prompt for class $c$ is constructed by concatenating the learnable prompt and a fixed class token. Specifically, let $\mathbf{p}_c \in \mathbb{R}^{N_t \times d_t}$ denote the class token for class $c$, and $\delta_k \in \mathbb{R}^{N_p \times d_t}$ denote the learnable prompt from client $k$, where $N_t$ and $N_p$ are the lengths of the fixed and learnable prompts, $d_t$ is the dimension of hidden embedding, respectively. The full prompt can be formed as $P_c^{(k)} = [\delta_k; \mathbf{p}_c]$. The text feature for class $c$ is computed by feeding the prompt into a text encoder $\mathcal{T}(\cdot)$: $\mathbf{f}_t^c = \mathcal{T}(P_c^{(k)})$. Given an input image $\mathbf{x}$, the image feature is extracted by the frozen image encoder $f$: $\mathbf{f}_v = f(\mathbf{x})$. The similarity score between the image and class-$c$ textual features is calculated as $\rho_c = \text{Cosine}(\mathbf{f}_t^c, \mathbf{f}_v)$, and the prediction probability is computed via softmax over similarities: $p(\hat{y} = c \mid \mathbf{x}) = \frac{\exp(\rho_c/\tau)}{\sum_{j=1}^K \exp(\rho_j/\tau)}$. The training objective is the cross-entropy loss between the similarity-based prediction and the ground-truth label: $\mathcal{L}_{\text{ce}} = \ell(\boldsymbol{\rho}, \mathbf{e}_y)$, where $\boldsymbol{\rho} = [\rho_1, \ldots, \rho_K]$, $\mathbf{e}_y$ is the one-hot label vector, and $\ell$ denotes the cross-entropy loss function. After local training, client $C_k$ uploads its learned prompt $\delta_k$ to the server. The server then aggregates the local prompts $\{\delta_k\}_{k=1}^K$ to construct a global prompt $\delta_g$, typically by weighted averaging: $\delta_g = \sum_{k=1}^K p_k \cdot \delta_k$.

# 4 The Proposed `GPR-NIAM` Method

In this section, we propose the non-interfering attention masking method for one-shot federated prompt learning, called `GPR-NIAM`. Its core idea is to preserve the transferable knowledge by reducing the influence of learnable prompt tokens on the input text tokens, as illustrated in Figure 2.

## 4.1 The Attention Isolation (AttnIso) Module

Prompt tuning typically focuses on fitting local training tasks. However, data heterogeneity often leads to divergent optimization across clients and weakens its cross-task generalization. To address this, the AttnIso module designs attention masks to selectively block the information flow from learnable prompt tokens to original text tokens (e.g. "a photo of a [CLASS]"). We denote the text attention matrix as $M \in \mathbb{R}^{(1+N_p+N_t)*(1+N_p+N_t)}$, where $N_p$ is the number of learnable prompts, $N_t$ is the length of input textual feature tokens. The additional 1 corresponds to the [EOS] token. In addition, we apply customized attention control for the [EOS] token to ensure proper aggregation of contextual information from both the prompt and the input tokens.

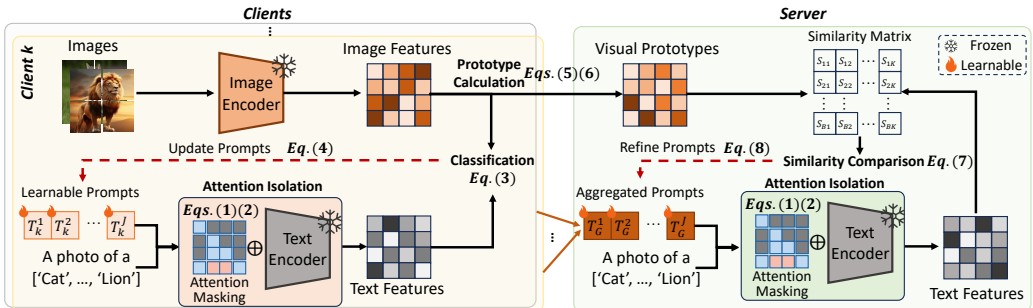

Figure 2: Illustration of the GPR-NIAM framework. The Attention Isolation module preserves the original text embeddings to reduce the forgetting of generalized knowledge. The global prompt is refined using visual prototypes from all clients to mitigate data heterogeneity.

To achieve the above goal, the text attention matrix $M$ can be formulated as follows,

$$
M^{i,j} = \begin{cases} -\infty, & \begin{aligned} &\text{if } i \in [1, N_p] \wedge j \in [N_p + 1, N_p + N_t] \\ &\text{or } i \in [N_p + 1, N_p + N_t] \wedge j \in [1, N_p] \\ &\text{or } j = N_p + N_t + 1 \end{aligned} \quad \text{(Hard Masking)} \\ \lambda, & \text{if } i = N_p + N_t + 1 \wedge j \in [N_p + 1, N_p + N_t] \quad \text{(Reweighting)} \\ 0, & \text{otherwise} \quad\quad\quad\quad\quad\quad\quad\quad\quad\quad\quad \text{(Full Attention)} \end{cases}
\tag{1}
$$

where $M^{i,j}$ denotes the $i$-th row and $j$-th column element of $M$, and $\wedge$ denotes logical AND. Moreover, to integrate the attention mask $M$ into the text encoder $\mathcal{T}$, we replace the original computation with the following:

$$
\mathbf{y}^{(l+1)} = \mathcal{T}_{l+1}(\mathbf{y}^{(l)}, M) = \text{Concat}(h_1, h_2, \ldots, h_H)W^O = [\tilde{p}_{l+1}^1, \ldots, \tilde{p}_{l+1}^{N_p}, \tilde{t}_{l+1}^1, \ldots, \tilde{t}_{l+1}^{N_t}, \tilde{s}_{l+1}^{\text{eos}}],
$$
$$
\text{where } h_i = \text{softmax}\left( \frac{Q_i K_i^T}{\sqrt{d_k}} + M \right) V_i, \quad i = 1, \ldots, H
\tag{2}
$$

where $\mathbf{y}^{(l)} \in \mathbb{R}^{(N_p + N_t + 1) \times d_t}$ and $\mathbf{y}^{(l+1)} \in \mathbb{R}^{(N_p + N_t + 1) \times d_t}$ are the input and output of the $(l+1)$-th encoder layer. $\tilde{p}_{l+1}^i, \tilde{t}_{l+1}^j$, and $\tilde{s}_{l+1}^{\text{eos}}$ represent the output embeddings of the $i$-th prompt token, the $j$-th text token, and the [EOS] token, respectively. $Q_i, K_i, V_i \in \mathbb{R}^{(N_p + N_t + 1) \times d_k}$ are the query, key, and value matrices for the $i$-th attention head, $d_k = d_t / H$ denotes the hidden dimension of each attention head. $\text{Concat}(h_1, \ldots, h_H) \in \mathbb{R}^{(N_p + N_t + 1) \times H d_k}$ denotes the concatenated outputs of all heads, and $W^O \in \mathbb{R}^{H d_k \times d_t}$ is the output projection matrix mapping the result back to $d_t$. Following the masked encoding process, we obtain the final output of the text encoder as: $\mathbf{y}^{(L)} = [\tilde{p}_L^1, \ldots, \tilde{p}_L^{N_p}, \tilde{t}_L^1, \ldots, \tilde{t}_L^{N_t}, \tilde{s}_L^{\text{eos}}]$. Afterwards, $\tilde{s}_L^{\text{eos}}$ is projected into a shared embedding space via a pre-trained head $H_t(\cdot)$. On the visual side, the input image $\mathbf{x}$ is encoded by a frozen vision encoder $f(\cdot)$ to obtain the visual representation. Overall, it focuses on minimizing the objective:

$$
\min_{\delta_1, \ldots, \delta_K} \sum_{k=1}^K \frac{|\mathcal{D}_k|}{|\mathcal{D}|} \cdot \mathbb{E}_{(\mathbf{x}, y) \sim \mathcal{D}_k} \left[ -\log\left( \frac{\exp\left(\text{sim}\left(f_v(\mathbf{x}), H_t\left(\tilde{s}_{L,k}^{(y)}\right)\right)\right)}{\sum_{c=1}^C \exp\left(\text{sim}\left(f_v(\mathbf{x}), H_t\left(\tilde{s}_{L,k}^{(c)}\right)\right)\right)} \right) \right], \tag{3}
$$

where $\tilde{s}_{L,k}^{(c)}$ denotes client $k$'s [EOS] embedding for class $c$, $\text{sim}(\cdot, \cdot)$ denotes cosine similarity function. The learnable prompts can be optimized by gradient descent in all clients:

$$
\delta_k \leftarrow \delta_k - \eta \cdot \frac{1}{|\mathcal{B}_k|} \sum_{(\mathbf{x}, y) \in \mathcal{B}_k} \nabla_{\delta_k} \left[ -\log\left( \frac{\exp\left(\text{sim}\left(f_v(\mathbf{x}), H_t\left(\tilde{s}_{L,k}^{(y)}\right)\right)\right)}{\sum_{c=1}^C \exp\left(\text{sim}\left(f_v(\mathbf{x}), H_t\left(\tilde{s}_{L,k}^{(c)}\right)\right)\right)} \right) \right], \quad \forall k \in [K], \tag{4}
$$

where $\eta$ denotes the learning rate, $\mathcal{B}_k$ is a batch data.

## 4.2 Cross-Silo Collaborative Refinement (CSCR) Module

To further alleviate the inconsistency caused by data heterogeneity, the CSCR module calibrates the global prompt through prototype-guided refinement across clients. To support this, each client $C_k$

extracts class-wise visual prototypes from its local dataset $\mathcal{D}_k$. For class $c$, the prototype is computed as a randomly weighted average of the feature vectors from local samples in that class:

$$f_{v,k}^c = \sum_{\mathbf{x} \in \mathcal{D}_k^c} \alpha_{\mathbf{x}} \cdot f_v(\mathbf{x}), \quad \text{where} \sum_{\mathbf{x} \in \mathcal{D}_k^c} \alpha_{\mathbf{x}} = 1, \ \alpha_{\mathbf{x}} \sim \mathcal{U}(0,1). \tag{5}$$

where $\alpha_{\mathbf{x}}$ denotes a random weight for each sample $\mathbf{x}$ in class $c$, drawn from a uniform distribution and normalized to ensure the weights sum to 1. Notably, we can randomly sample the weight vector $n$ times to generate multiple diverse prototypes per class $\mathbf{P}_k^c = \left[ f_{v,k}^{c,1}, f_{v,k}^{c,2}, \ldots, f_{v,k}^{c,n} \right]$, which enhances representational diversity, which differs from existing methods [47, 48, 49, 50, 51]. $f_{v,k}^{c,i}$ denotes $i$-th visual prototypes of class $c$ in client $k$. Afterwards, each client $C_k$ sends $\mathbf{P}_k = \{\mathbf{P}_k^1, \ldots, \mathbf{P}_k^N\}$ to the server, which aggregates all received prototypes into a global pool $\mathcal{P}$:

$$\mathcal{P} = \bigcup_{k=1}^{K} \mathbf{P}_k = \{\mathbf{P}_k^c \mid k \in [K], \ c \in [n]\}, \tag{6}$$

To calibrate the global prompt $\delta_g$, we leverage the global prototype pool $\mathcal{P} = \{\mathcal{P}^1, \ldots, \mathcal{P}^C\}$ and enforce the alignment between textual representations and visual prototypes via a class-level contrastive objective. Specifically, we design the following loss function:

$$\min_{\delta_g} -\frac{1}{|\mathcal{P}|} \sum_{f \in \mathcal{P}} \log \sigma \left( \text{sim}\left(f, \ H_t(\mathcal{T}(\delta_g, \mathbf{t}_c))\right) - \log \sum_{j=1}^C \exp\left(\text{sim}\left(f, \ H_t(\mathcal{T}(\delta_g, \mathbf{t}_j))\right)\right)\right), \tag{7}$$

where $\sigma(\cdot)$ is the sigmoid function, $\mathbf{t}_c$ denotes 'a photo of c-th classname'. We then follow the gradient descent method to obtain the optimal parameters:

$$\delta_g \leftarrow \delta_g - \eta \cdot \frac{1}{|\mathcal{P}|} \sum_{(f,c) \in \mathcal{P}} \nabla_{\delta_g} \log \sigma \left( \text{sim}\left(f, \ H_t(\mathcal{T}(\delta_g, \mathbf{t}_c))\right) - \log \sum_{j=1}^C \exp\left(\text{sim}\left(f, \ H_t(\mathcal{T}(\delta_g, \mathbf{t}_j))\right)\right)\right), \tag{8}$$

where the global prompt $\delta_g$ is initialized as the weighted average of all client-specific prompts (i.e., $\delta_g \leftarrow \sum_{k=1}^K \frac{|\mathcal{D}_k|}{|\mathcal{D}|} \cdot \delta_k$), and $\eta$ is a learning rate.

### 4.3 Training Procedure

In this section, we provide the detailed training procedure of GPR-NIAM in Algorithm 1. As a two-stage optimization strategy, GPR-NIAM consists of a local prompt tuning phase and a global prompt refinement phase. Notably, it can be easily integrated into multiple prompt learning methods such as CoOp and TCP. In the first stage, each client initializes its local prompt from the global prompt and updates it on private data with masked attention learning. Simultaneously, class-wise visual prototypes are obtained by performing randomly weighted aggregation over local feature representations. Subsequently, the global prompt is optimized to align with all class prototypes in this pool via cross-silo collaborative retraining. With the two-stage design, GPR-NIAM alleviates the challenges posed by heterogeneous client data and promotes the retention of transferable knowledge for unseen tasks.

Moreover, the local prompt tuning stage is optional. Firstly, this stage provides a more meaningful initialization for the global prompt by adapting it to all clients data. Secondly, it generates class-wise visual prototypes to guide global refinement. While the framework can still perform global refinement without this phase, since it does not affect prototype modeling, omitting it might result in less effective initialization and slightly limit the overall refinement performance.

## 5 Experimental Evaluation

### 5.1 Experiment Settings

**Datasets** Following existing works [17, 52], experiments are conducted on CIFAR10 [53], Oxford-Pets [54], Caltech101 [55], DTD [56], FGVCAircraft [57], Flowers102 [58], StanfordCars [59], and UCF101 [60] to evaluate base-to-base/novel generalization, and on Office-Home [61] and DomainNet [62] to assess leave-one-domain-out generalization. Their statistics are shown in Table 5.

**Algorithm 1** GPR-NIAM

1: **Initialize** the global prompt parameter $\delta^0$
2: **// Local Prompt Tuning Stage**
3: **for** each client $C_k \in \mathcal{C}$ in parallel **do**
4:     Initialize local prompt parameter $\delta_k = \delta^0$
5:     **for** $e = 1, \ldots, E$ **do**
6:         Sample a mini-batch $\zeta$ from local data $\mathcal{D}_k$
7:         Update local prompt $\delta_k$ using masked attention by Eq. (4)
8:     **end for**
9:     Compute visual prototype features for each class using Eq. (5)
10: **end for**
11: **// Global Prompt Refinement Stage**
12: Aggregate all visual prototypes to form a global prototype pool $\mathcal{P}$ by Eq. (6)
13: **for** $e' = 1, \ldots, E'$ **do**
14:     Sample a mini-batch of prototypes $\zeta'$ from $\mathcal{P}$
15:     Update global prompt $\delta_g$ via prototype-guided alignment using Eq. (8)
16: **end for**

**Evaluation Metrics**    Following studies [47, 49, 63], we report the Top-1 Accuracy (Acc) results. For base-to-base/novel generalization tasks, we also compute the harmonic mean (HM $= \frac{2 \cdot \text{Acc}_{\text{base}} \cdot \text{Acc}_{\text{novel}}}{\text{Acc}_{\text{base}} + \text{Acc}_{\text{novel}}}$) of accuracy on base ($\text{Acc}_{\text{base}}$) and novel classes ($\text{Acc}_{\text{novel}}$) to evaluate overall performance.

**Implementation Details**    In all experiments, we set local training and global refinement epochs to 10, using the SGD optimizer with a learning rate of 0.001, a weight decay of 0.001, and a batch size of

Table 1: Performance comparison between GPR-NIAM and baselines on 8 datasets with $\beta = 0.5$ in the base-to-base/novel generalization cases.

| (a) Average over 8 datasets | | | | (b) CIFAR10 | | | | (c) OxfordPets | | | |
|---|---|---|---|---|---|---|---|---|---|---|---|
| **Method** | **Base** | **Novel** | **HM** | **Method** | **Base** | **Novel** | **HM** | **Method** | **Base** | **Novel** | **HM** |
| CLIP | 65.79 | 74.32 | 69.70 | CLIP | 95.30 | 95.50 | 95.39 | CLIP | 81.53 | 95.33 | 87.89 |
| PromptFL | 72.57 | 72.80 | 72.64 | PromptFL | 95.14 | 96.26 | 95.69 | PromptFL | 91.31 | 93.70 | 92.48 |
| RPO | 70.52 | 73.63 | 71.99 | RPO | 95.68 | 94.74 | 95.20 | RPO | 89.00 | 95.49 | 92.13 |
| FedTPG | 70.46 | 73.09 | 71.69 | FedTPG | 95.38 | 96.56 | 95.96 | FedTPG | 91.66 | 95.13 | 93.36 |
| TCP | 72.46 | 74.60 | 73.41 | TCP | 95.68 | 96.52 | 96.09 | TCP | 90.01 | 95.39 | 92.62 |
| PromptFolio | 73.25 | 73.35 | 73.15 | PromptFolio | **96.04** | 96.16 | 96.06 | PromptFolio | 91.13 | **96.10** | 93.54 |
| DP-FPL | 71.90 | 71.75 | 69.88 | DP-FPL | 95.42 | 95.18 | 95.29 | DP-FPL | 91.97 | 94.80 | 93.36 |
| FedDISC | 56.43 | 54.84 | 53.87 | FedDISC | 94.30 | 92.82 | 93.55 | FedDISC | 83.83 | 72.07 | 77.50 |
| FedELMY | **76.62** | 48.23 | 60.03 | FedELMY | 95.32 | 40.00 | 79.66 | FedELMY | 76.47 | 76.47 | 82.38 |
| GPR-NIAM$_P$ | 75.56 | 73.28 | 74.24 | GPR-NIAM$_P$ | 95.66 | **97.00** | **96.32** | GPR-NIAM$_P$ | 91.18 | 95.89 | 93.47 |
| GPR-NIAM$_T$ | 76.43 | **74.48** | **75.30** | GPR-NIAM$_T$ | 95.84 | 96.48 | 96.15 | GPR-NIAM$_T$ | **92.48** | 95.14 | **93.79** |

| (d) Caltech101 | | | | (e) DTD | | | | (f) FGVCAircraft | | | |
|---|---|---|---|---|---|---|---|---|---|---|---|
| **Method** | **Base** | **Novel** | **HM** | **Method** | **Base** | **Novel** | **HM** | **Method** | **Base** | **Novel** | **HM** |
| CLIP | 82.27 | 94.05 | 87.77 | CLIP | 53.35 | **59.05** | 56.06 | CLIP | 26.51 | 31.21 | 28.67 |
| PromptFL | 88.27 | 93.88 | 90.99 | PromptFL | 60.06 | 53.86 | 56.79 | PromptFL | 31.19 | 30.91 | 31.05 |
| RPO | 89.90 | 93.80 | 91.81 | RPO | 54.62 | 59.17 | 56.81 | RPO | 29.27 | 30.43 | 29.84 |
| FedTPG | 89.12 | 92.61 | 90.83 | FedTPG | 56.71 | 55.43 | 56.06 | FedTPG | 29.57 | 30.25 | 29.90 |
| TCP | 88.20 | **95.33** | 91.62 | TCP | 61.68 | 55.48 | 58.41 | TCP | 30.77 | **32.71** | 31.71 |
| PromptFolio | 87.28 | 93.12 | 90.10 | PromptFolio | 61.11 | 54.58 | 57.66 | PromptFolio | 31.37 | 32.39 | **31.87** |
| DP-FPL | 87.92 | 93.46 | 90.60 | DP-FPL | 62.26 | 49.63 | 55.24 | DP-FPL | 28.85 | 28.15 | 28.49 |
| FedDISC | 44.75 | 85.22 | 58.68 | FedDISC | **71.52** | 38.28 | 49.87 | FedDISC | 20.87 | 13.20 | 16.17 |
| FedELMY | 87.71 | 89.21 | 88.45 | FedELMY | 70.13 | 16.18 | 26.29 | FedELMY | **33.77** | 19.48 | 24.70 |
| GPR-NIAM$_P$ | 91.59 | 94.22 | 92.89 | GPR-NIAM$_P$ | 68.40 | 53.50 | 60.04 | GPR-NIAM$_P$ | 30.95 | 29.41 | 30.16 |
| GPR-NIAM$_T$ | **92.65** | 94.73 | **93.68** | GPR-NIAM$_T$ | 69.90 | 55.91 | **62.13** | GPR-NIAM$_T$ | 32.15 | 31.12 | 31.62 |

| (g) Flowers102 | | | | (h) StanfordCars | | | | (i) UCF101 | | | |
|---|---|---|---|---|---|---|---|---|---|---|---|
| **Method** | **Base** | **Novel** | **HM** | **Method** | **Base** | **Novel** | **HM** | **Method** | **Base** | **Novel** | **HM** |
| CLIP | 62.39 | 78.10 | 69.36 | CLIP | 56.87 | 69.40 | 62.51 | CLIP | 68.12 | 71.92 | 69.97 |
| PromptFL | 79.88 | 75.15 | 77.44 | PromptFL | 62.96 | 67.30 | 65.02 | PromptFL | 71.82 | 71.55 | 71.68 |
| RPO | 74.92 | 76.21 | 75.56 | RPO | 58.64 | 68.73 | 63.29 | RPO | 72.19 | 70.49 | 71.33 |
| FedTPG | 72.30 | 77.26 | 74.70 | FedTPG | 58.67 | 69.78 | 63.74 | FedTPG | 70.34 | 67.74 | 69.01 |
| TCP | 77.25 | **78.73** | 77.98 | TCP | 60.19 | 71.18 | 65.22 | TCP | 75.93 | 71.52 | 73.65 |
| PromptFolio | 81.04 | 74.51 | 77.63 | PromptFolio | 61.22 | **69.91** | 65.27 | PromptFolio | 76.88 | 69.75 | 73.14 |
| DP-FPL | 74.63 | 77.26 | 75.92 | DP-FPL | 59.07 | 66.05 | 62.36 | DP-FPL | 75.14 | 69.49 | 72.20 |
| FedDISC | 24.95 | 32.21 | 28.11 | FedDISC | 42.77 | 49.92 | 46.07 | FedDISC | 68.49 | 55.03 | 61.03 |
| FedELMY | **89.79** | 55.15 | 68.33 | FedELMY | **67.09** | 47.97 | 55.94 | FedELMY | **79.94** | 41.41 | 54.56 |
| GPR-NIAM$_P$ | 84.54 | 78.52 | 81.42 | GPR-NIAM$_P$ | 63.59 | 67.48 | 65.48 | GPR-NIAM$_P$ | 78.57 | 70.28 | 74.19 |
| GPR-NIAM$_T$ | 86.88 | 77.05 | **81.67** | GPR-NIAM$_T$ | 63.54 | 69.18 | **66.24** | GPR-NIAM$_T$ | 78.04 | **76.27** | **77.14** |

Table 2: Performance comparison on Office-Home and DomainNet under the leave-one-domain-out setting, with each domain's data randomly split across 3 clients.

| Method | Office-Home | | | | | DomainNet | | | | | | |
|--------|------|---------|---------|-----------|---------|---------|-----------|----------|-----------|-------|--------|---------|
| | Art | Clipart | Product | RealWorld | Average | Clipart | Infograph | Painting | Quickdraw | Real | Sketch | Average |
| CLIP | 74.57 | 62.33 | 80.17 | 79.38 | 74.08 | 92.87 | 77.02 | 85.35 | 43.06 | 98.61 | 94.22 | 81.85 |
| PromptFL | 74.24 | 66.20 | 82.47 | 79.77 | 75.17 | 93.87 | 80.82 | 87.76 | 49.47 | 98.72 | 94.82 | 84.24 |
| RPO | 74.86 | 64.33 | 84.31 | 80.74 | 76.06 | 93.53 | 78.32 | 86.15 | 49.35 | 98.83 | 94.87 | 83.50 |
| FedTPG | 74.04 | 64.55 | 83.32 | 80.01 | 75.48 | **95.14** | 80.93 | 88.25 | 52.89 | 98.97 | 95.40 | 85.26 |
| TCP | 75.31 | 66.20 | 83.17 | 80.97 | 76.41 | 94.13 | 79.80 | 88.05 | 45.85 | 98.80 | 95.34 | 83.66 |
| PromptFolio | 77.21 | 67.74 | 84.74 | 81.98 | 77.92 | 94.87 | **81.19** | 88.61 | 47.59 | 98.85 | 95.02 | 84.35 |
| DP-FPL | 75.03 | 65.91 | 83.71 | 79.82 | 76.12 | 95.03 | 78.41 | 86.74 | 54.03 | 98.83 | 94.85 | 84.64 |
| FedDISC | 72.27 | 65.40 | 81.77 | 79.11 | 74.63 | 93.82 | 80.42 | 86.17 | 43.74 | 98.66 | 94.05 | 82.81 |
| FedELMY | 75.11 | 66.34 | 84.12 | 79.69 | 76.31 | 94.21 | 77.65 | 86.81 | 48.78 | 98.83 | 94.88 | 83.53 |
| GPR-NIAM$_P$ | 78.32 | 71.40 | 87.61 | 84.96 | 80.57 | 94.72 | 80.13 | **88.70** | **56.98** | **99.18** | 94.88 | 85.76 |
| GPR-NIAM$_T$ | **79.39** | **72.06** | **88.03** | **85.49** | **81.24** | 95.03 | 80.82 | 87.92 | 56.32 | 99.08 | **95.60** | **85.79** |

32. The number of communication rounds is 1. We simulate 50 clients for CIFAR-10 and 10 clients for other datasets in base-to-base/novel generalization, while using 9 clients for Office-Home and 15 for DomainNet. To simulate non-IID data, we adopt a Dirichlet distribution with parameter $\beta = 0.5$. Each client generates $n \in \{5, 10, 20\}$ visual prototypes per class, and the reweighting parameter $\lambda$ is selected from $\{0.2, 0.5, 0.7, 1.0\}$. More details and results can be found in the Appendix. And each client has one NVIDIA RTX 3090 with 24 GB GPU for training.

## 5.2 Results and Discussion

This section compares GPR-NIAM-enhanced variants (GPR-NIAM$_P$ and GPR-NIAM$_T$, derived by integrating GPR-NIAM into PromptFL and TCP, respectively) with nine baselines: Zero-Shot CLIP [10] with hand-crafted text prompt template, e.g., "a photo of a [class]" [64], PromptFL [65], RPO [66], FedTPG [52], TCP [67], PromptFolio [17], DP-FPL [22], FedDISC [28], FedELMY [42]. For fair comparison, all methods use the same backbone: a frozen CLIP with ViT-B16 [10]. Following prior work, the learnable prompt is a $10 \times 512$ parameter matrix, with 10 prompt tokens and a 512-dimensional embedding.

**Base-to-Base/Novel Generalization.** We evaluate the generalization capability of GPR-NIAM by comparing its performance on base and novel classes, and the overall harmonic mean (HM) across all benchmarks. The results are presented in Table 1 and key observations are summarized as follows:

- GPR-NIAM attains second-best performance on base classes among all methods, showing particular competitiveness against FedMLEY while maintaining more balanced capabilities across all classes.

- GPR-NIAM yields competitive performance on novel classes. Notably, it ranks first on Flowers102 and UCF101, underscoring its strong generalization capability to unseen tasks. Moreover, FedDISC may fail on both base and novel classes due to inevitable gaps between synthetic and real samples.

- Benefiting from the synergy of attention masking and cross-silo collaborative retraining, GPR-NIAM achieves the best HM scores across all benchmarks.

**Leave-One-Domain-Out Generalization.** Table 2 presents the domain generalization results under a leave-one-domain-out setting. The main findings are summarized below:

- GPR-NIAM consistently achieve the highest average accuracy across most domains. Notably, GPR-NIAM$_T$ ranks first in 9 out of 12 domains, highlighting its strength in cross-domain generalization, especially in challenging cases like Infograph and Quickdraw.

- By combining token-level and client-level perspectives, GPR-NIAM outperforms single-view methods such as PromptFL and FedTPG, which benefits from attention masking for preserving original semantics and a unified global prompt for cross-domain knowledge integration.

## 5.3 Ablation Study

This section presents an ablation study to examine the effects of key components in GPR-NIAM, including the Attention Isolation (AttnIso) module and the Cross-Silo Collaborative Retraining (CSCR) module. The results are reported in Table 3.

Table 3: Ablation results averaged over 8 datasets (base-to-base/novel generalization) and across all domains (leave-one-domain-out generalization).

| Method | Base to Base/Novel Generalization | | | Leave-One-Domain-Out Generalization | |
|---|---|---|---|---|---|
| | Base | Novel | HM | Office-Home | DomainNet |
| PromptFL | 64.06 | 71.48 | 66.98 | 75.17 | 84.24 |
| + AttnIso$_{HA}$ | 62.19 | **72.27** | 67.04 | 77.26 | 84.87 |
| + AttnIso$_{RE}$ | 64.22 | 71.51 | 67.76 | 75.67 | 84.36 |
| + AttnIso$_{HA+RE}$ | 64.11 | 72.66 | 67.87 | 77.83 | 85.01 |
| + CSCR | **73.34** | 64.89 | 67.24 | 78.59 | 85.11 |
| + AttnIso$_{HA}$ + CSCR | 69.32 | 70.78 | 70.06 | 79.13 | 85.46 |
| + AttnIso$_{RE}$ + CSCR | 70.56 | 68.11 | 69.65 | 78.42 | 85.23 |
| + AttnIso$_{HA+RE}$ + CSCR | 72.32 | 69.47 | **70.72** | **80.57** | **85.76** |

- Different configurations of the AttnIso module exhibit distinct strengths. Hard masking (AttnIso$_{HA}$) effectively suppresses token interference and enhances novel-class generalization, while reweighting (AttnIso$_{RE}$) provides a more stable improvement. Their combination leads to improved overall generalization, showing that the two strategies are complementary.

- The CSCR module plays a vital role in optimizing base-class performance by learning a unified prompt that mitigates inter-client discrepancies, which can alleviate the negative impact of data heterogeneity. However, its contribution to novel-class generalization remains limited due to insufficient adaptability to unseen categories.

- By integrating both modules, `GPR-NIAM` achieves an optimal balance between acquiring new knowledge and preserving generalization. Importantly, this also suggests a promising direction for future work by combining complementary strategies rather than relying on a single one.

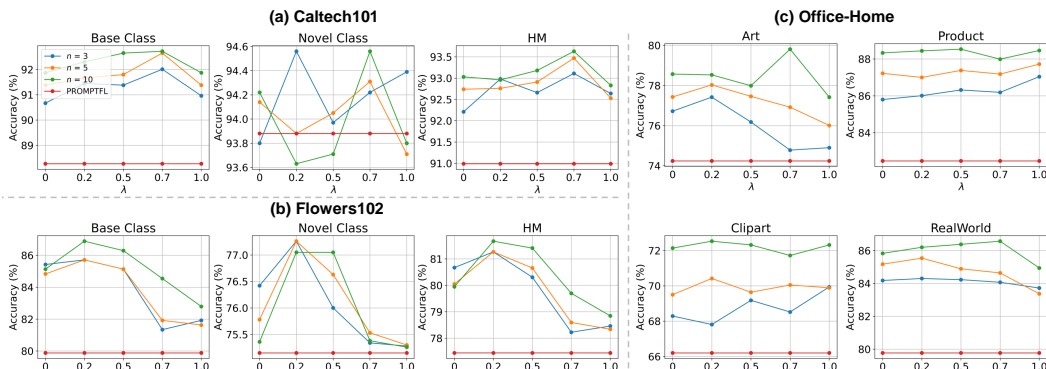

Figure 3: The impact of hyperparameters on performance is evaluated by varying $\lambda$ from $\{0, 0.2, 0.5, 0.7, 1.0\}$, the number of prototypes per class $n$ from $\{3, 5, 10\}$ on three datasets.

## 5.4 Sensitivity Analysis

This section evaluates the robustness of `GPR-NIAM` across varying hyperparameter settings on the DTD, Flowers102 and Office-Home datasets. Specifically, we investigate the effects of hyperparameters $n$ (the number of prototypes generated per class on the client side) and $\lambda$ (the weighting factor in the AttnIso module), evaluated over $\{3, 5, 10\}$ and $\{0, 0.2, 0.5, 0.7, 1.0\}$, respectively. As depicted in Figure 3, `GPR-NIAM` consistently achieves higher overall HM scores than its baselines in all cases of the base-to-base/novel generalization task, showcasing its superior balance between retaining transferable knowledge and learning task-specific knowledge. On the one hand, `GPR-NIAM` demonstrates consistent improvements over baselines on base classes, which benefits from the CSCR module's capacity to alleviate data heterogeneity. On the other hand `GPR-NIAM` exhibits competitive performance on novel-class generalization, frequently outperforming other methods, which is attributed to the attention masking mechanism of the AttnIso module. Furthermore, in the Leave-One-Domain-Out Generalization task, similar conclusions are drawn, as `GPR-NIAM` outperforms the baseline across multiple domains.

Table 4: Performance comparison between GPR-NIAM and baselines on DTD and Flowers102 with local training epochs $\in \{1, 5, 20\}$ in the base-to-base/novel generalization setting.

| (a) DTD (Epochs=1) | | | | (b) DTD (Epochs=5) | | | | (c) DTD (Epochs=20) | | | |
|---|---|---|---|---|---|---|---|---|---|---|---|
| **Method** | **Base** | **Novel** | **HM** | **Method** | **Base** | **Novel** | **HM** | **Method** | **Base** | **Novel** | **HM** |
| CLIP | 53.35 | 59.05 | 56.06 | CLIP | 53.35 | 59.05 | 56.06 | CLIP | 53.35 | 59.05 | 56.06 |
| PromptFL | 50.34 | 51.69 | 51.01 | PromptFL | 55.55 | 56.76 | 56.15 | PromptFL | 63.31 | 51.08 | 56.54 |
| RPO | 52.77 | **59.42** | 55.90 | RPO | 53.24 | **59.78** | 56.32 | RPO | 53.93 | **60.26** | 56.92 |
| FedTPG | 40.27 | 36.47 | 38.28 | FedTPG | 49.88 | 47.22 | 48.51 | FedTPG | 56.25 | 56.28 | 56.26 |
| TCP | 53.24 | 53.38 | 53.31 | TCP | 57.63 | 56.40 | 57.01 | TCP | 63.07 | 59.54 | 61.25 |
| PromptFolio | 52.54 | 53.50 | 53.02 | PromptFolio | 58.68 | 56.28 | 57.45 | PromptFolio | 58.68 | 57.60 | 58.13 |
| DP-FPL | 43.75 | 41.54 | 42.61 | DP-FPL | 59.25 | 50.12 | 54.30 | DP-FPL | 64.12 | 47.82 | 54.78 |
| GPR-NIAM$_P$ | **65.97** | 51.20 | **57.65** | GPR-NIAM$_P$ | **68.51** | 52.65 | 59.54 | GPR-NIAM$_P$ | **70.13** | 53.26 | 60.54 |
| GPR-NIAM$_T$ | 63.07 | 51.81 | 56.89 | GPR-NIAM$_T$ | 65.85 | 55.07 | **59.98** | GPR-NIAM$_T$ | 69.67 | 54.83 | **61.36** |

| (d) Flowers102 (Epochs=1) | | | | (e) Flowers102 (Epochs=5) | | | | (f) Flowers102 (Epochs=20) | | | |
|---|---|---|---|---|---|---|---|---|---|---|---|
| **Method** | **Base** | **Novel** | **HM** | **Method** | **Base** | **Novel** | **HM** | **Method** | **Base** | **Novel** | **HM** |
| CLIP | 62.39 | 78.10 | 69.36 | CLIP | 62.39 | 78.10 | 69.36 | CLIP | 62.39 | 78.10 | 69.36 |
| PromptFL | 70.84 | 74.10 | 72.43 | PromptFL | 78.42 | 76.00 | 77.19 | PromptFL | 80.46 | 73.68 | 76.92 |
| RPO | 69.97 | 76.21 | 72.95 | RPO | 71.72 | 76.84 | 74.19 | RPO | 74.05 | 77.05 | 75.52 |
| FedTPG | 68.51 | 77.05 | 72.53 | FedTPG | 73.76 | 77.05 | 75.37 | FedTPG | 74.34 | 77.05 | 75.67 |
| TCP | 75.51 | 78.10 | 76.78 | TCP | 76.09 | 79.15 | 77.59 | TCP | 79.88 | 78.31 | 79.09 |
| PromptFolio | 73.76 | 76.42 | 75.06 | PromptFolio | 79.59 | 75.57 | 77.53 | PromptFolio | 83.09 | 74.52 | 78.57 |
| DP-FPL | 69.38 | **78.31** | 73.58 | DP-FPL | 73.76 | **79.57** | 76.55 | DP-FPL | 75.80 | 75.78 | 75.79 |
| GPR-NIAM$_P$ | 82.50 | 75.57 | 78.89 | GPR-NIAM$_P$ | 84.83 | 78.31 | 81.44 | GPR-NIAM$_P$ | 85.13 | 77.47 | 81.12 |
| GPR-NIAM$_T$ | 87.11 | 77.83 | **82.20** | GPR-NIAM$_T$ | **86.58** | 78.94 | **82.59** | GPR-NIAM$_T$ | **87.75** | **78.73** | **83.00** |

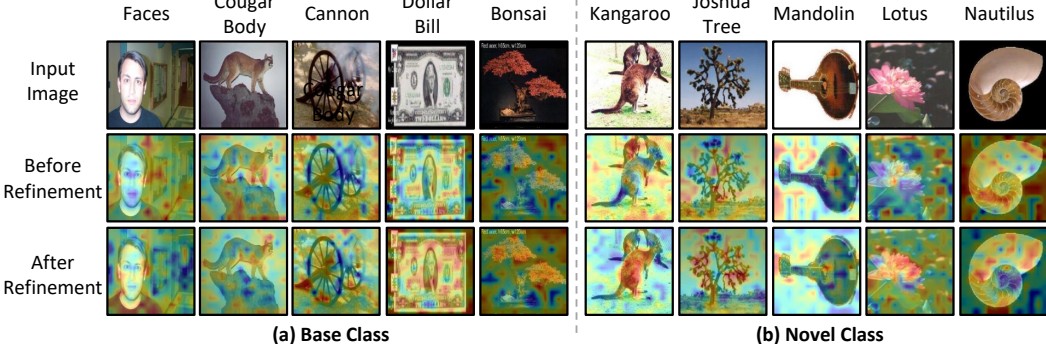

Figure 4: Visualization of similarity heatmaps between text features and image patches before and after prompt refinement on Caltech101. In the similarity maps, red and yellow indicate higher similarity between image patches and the text features, while blue represents lower similarity.

## 5.5 Performance Comparison across Different Local Training Epoch Settings

To assess the robustness of the proposed GPR-NIAM under different local training efforts, we evaluate its performance with local epochs $\in 1, 5, 20$, a data heterogeneity level of $\beta = 0.5$, and 10 participating clients. As shown in Table 4, both GPR-NIAM$_P$ and GPR-NIAM$_T$ consistently outperform baselines across all datasets and settings. Notably, GPR-NIAM$_T$ achieves the highest harmonic mean (HM) in most cases, indicating better generalization across base and novel classes. On the Flowers102 dataset, it maintains strong performance with increasing local epochs, reflecting robustness against overfitting. Overall, both variants remain stable and effective despite changing local training dynamics.

## 5.6 Effectiveness of Global Prompt Refinement

This section aims to evaluate the effectiveness of the global prompt refinement method. Specifically, we visualize the similarity maps between text features and image patches before and after refinement, as shown in Figure 4. Intuitively, the similarity responses between visual targets and textual semantics

are often scattered and misaligned before refinement. For example, in the "Dollar Bill" and "Bonsai" samples, the high similarity responses in the second row primarily concentrate on background regions near the image boundaries. In contrast, the responses become significantly more concentrated on semantically relevant regions, such as the face on the bill or the foliage of the bonsai after refinement. Notably, this advantage is also evident in novel classes. For instance, the refined prompts enable the model to better localize foreground objects while suppressing irrelevant background activations in "Kangaroo" and "Nautilus" samples. Moreover, it becomes tightly aligned with the spiral structure of the shell in the "Nautilus", indicating enhanced semantic grounding.

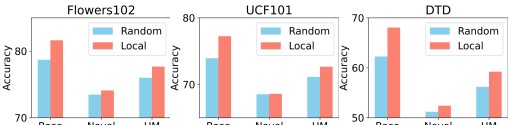

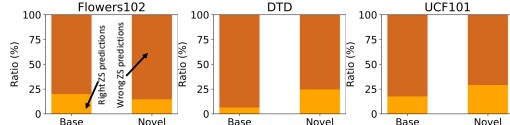

Figure 5: Comparison between Random and Local Prompt-Tuning for Initialization.

Figure 6: Failure case analysis. Most `GPR-NIAM` errors overlap with zero-shot CLIP.

## 5.7 Evaluation of Local Prompt Tuning as Initialization

This section evaluates the performance of random initialization and local prompt tuning (LPT) on Flowers102, UCF101 and DTD datasets. As shown in Figure 5, LPT yields superior performance compared to random initialization on all cases. This can be attributed to the local training that can be more aligned with the target data, allowing the model to better capture domain-specific features. In particular, LPT outperforms the random initialization by 6% on the DTD dataset for base classes, which highlights the method's ability to adapt more efficiently to the nuances of the texture.

## 5.8 Failure Case Analysis

This section reports the percentage of cases where the zero-shot CLIP model also failed in instances where the `GPR-NIAM` model made incorrect predictions (Figure 6). Clearly, there is a high overlap in the failure cases of both models. This reflects that the imprecision of zero-shot generalized knowledge hinders the model's generalization, which also provides insights for future work to address the challenges posed by the lack of domain-specific adaptation in generalization models.

# 6 Conclusions and Future Work

In this paper, we balance the model capability to learn task-specific knowledge for downstream tasks with its capacity to retain generalized knowledge. Specifically, we propose `GPR-NIAM` for one-shot FPL, which introduces an attention masking mechanism to limit the interaction between the learnable prompt tokens and the original text tokens. Furthermore, it learns a unified prompt through global prompt refinement, thereby mitigating the data heterogeneity issue. Experimental results demonstrate that `GPR-NIAM` significantly improves the performance of global prompts across diverse visual tasks.

Despite the promising results of `GPR-NIAM`, there remains room for improvement. The communication overhead from prototype transmission may become a bottleneck in large-scale federated systems with limited bandwidth, which we plan to address in future work. And both the multi-label [68, 69], multi-modal [70, 71, 72, 73] problems and real-world medical imaging tasks [74, 75, 76] become considerably more challenging in federated settings. In addition, several issues discussed in Section 5.8 remain open and deserve further investigation.

# Acknowledgments

This work is supported in part by the Key Research and Development Program of Shandong Province (Grant No. 2024TSGC0667); the Ministry of Education, Singapore, under its Academic Research Fund Tier 1 (RG101/24); the RIE2025 Industry Alignment Fund – Industry Collaboration Projects (IAF-ICP) (Award I2301E0026), administered by A*STAR, as well as supported by Alibaba Group and NTU Singapore through Alibaba-NTU Global e-Sustainability CorpLab (ANGEL).

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

# A Appendix

## A.1 Datasets

To conduct our evaluation, we selected 10 diverse visual classification datasets as benchmarks. Table 5 provides a detailed overview, including the number of classes, the size of training and testing sets, the number of domains, and the corresponding federated settings.

For datasets with multiple domains, we adopt the commonly used Office-Home benchmark, which includes four domains: Art, Clipart, Product, and RealWorld. In addition, we utilize DomainNet, a large-scale dataset spanning six domains: Clipart, Infograph, Painting, Quickdraw, Real, and Sketch. We focus on training with a selected subset of 20 classes from each dataset. Figure 7 illustrates representative raw instances from the two multi-domain datasets.

Table 5: Statistics and federated settings of the datasets used in our experiments.

| Dataset | Classes | Train | Test | | Domains | Federated Settings | |
| --- | --- | --- | --- | --- | --- | --- | --- |
| | | | Base | Novel | | Clients | Heterogeneity |
| CIFAR10 | 10 | 25,000 | 5,000 | 5,000 | 1 | 5/10/20 | 0.1/0.5 |
| OxfordPets | 37 | 1,785 | 1,782 | 1,887 | 1 | 5/10/20 | 0.1/0.5 |
| Caltech101 | 100 | 3,310 | 1,416 | 1,178 | 1 | 5/10/20 | 0.1/0.5 |
| DTD | 47 | 1,440 | 864 | 828 | 1 | 5/10/20 | 0.1/0.5 |
| FGVCAircraft | 100 | 3,333 | 1,667 | 1,666 | 1 | 5/10/20 | 0.1/0.5 |
| Flowers102 | 102 | 2,807 | 343 | 475 | 1 | 5/10/20 | 0.1/0.5 |
| StanfordCars | 196 | 4,052 | 4,002 | 3,998 | 1 | 5/10/20 | 0.1/0.5 |
| UCF101 | 101 | 3,926 | 1,895 | 1,888 | 1 | 5/10/20 | 0.1/0.5 |
| DomainNet | 20 | 30,777 | | | 6 | 9/15 | – |
| Office-Home | 65 | 15,588 | | | 4 | 15/25 | – |

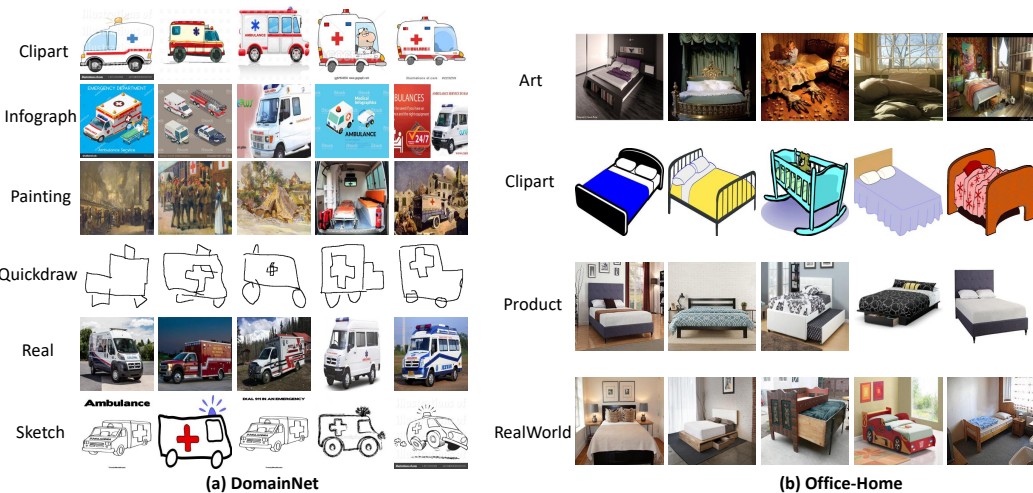

(a) DomainNet        (b) Office-Home

Figure 7: Illustrative examples of class instances across domains: "Ambulance" from DomainNet and "Bed" from Office-Home.

## A.2 Experimental Results

### A.2.1 Base-to-Base/Novel Generalization under High-Level Data Heterogeneity

To more comprehensively evaluate the effectiveness of the methods, we test all approaches on eight datasets under a higher level of heterogeneity ($\beta = 0.1$), focusing on their performance in the base-to-base/novel generalization setting. As presented in Table 6, our proposed method GPR-NIAM obtains the highest average

Table 6: Performance comparison between GPR-NIAM and baselines on 8 datasets with $\beta = 0.1$ in the base-to-base/novel generalization setting. The best performance is indicated in bold.

| (a) Average over 8 datasets | | | (b) CIFAR10 | | | (c) OxfordPets | | |
|---|---|---|---|---|---|---|---|---|
| **Method** | **Base** | **Novel** | **HM** | **Method** | **Base** | **Novel** | **HM** | **Method** | **Base** | **Novel** | **HM** |
| CLIP | 65.79 | 74.32 | 69.70 | CLIP | 95.30 | 95.50 | 95.39 | CLIP | 81.53 | 95.33 | 87.89 |
| PromptFL | 69.15 | 72.50 | 70.71 | PromptFL | 94.72 | 95.68 | 95.19 | PromptFL | 89.33 | 95.91 | 92.51 |
| RPO | 67.81 | 73.66 | 70.57 | RPO | 95.10 | 94.76 | 94.92 | RPO | 86.19 | 95.54 | 90.63 |
| FedTPG | 67.06 | 71.71 | 69.20 | FedTPG | 94.30 | 96.64 | 95.45 | FedTPG | 86.58 | 94.70 | 90.46 |
| TCP | 70.30 | 74.19 | 72.13 | TCP | 95.30 | 95.66 | 95.47 | TCP | 90.01 | **96.23** | 93.02 |
| PromptFolio | 69.49 | 73.44 | 71.33 | PromptFolio | 95.20 | 96.74 | 95.96 | PromptFolio | 90.90 | 95.17 | 92.99 |
| DP-FPL | 71.90 | 71.75 | 69.88 | DP-FPL | 94.24 | 95.42 | 94.82 | DP-FPL | 91.30 | 95.12 | **93.17** |
| FedDISC | 52.27 | 54.40 | 50.47 | FedDISC | 92.47 | 91.37 | 91.91 | FedDISC | 81.48 | 66.74 | 73.38 |
| FedELMY | 68.05 | 47.29 | 53.41 | FedELMY | 89.84 | 28.90 | 43.73 | FedELMY | 86.70 | 82.77 | 84.69 |
| GPR-NIAM$_P$ | **73.87** | 72.84 | 73.18 | GPR-NIAM$_P$ | 95.88 | **96.84** | **96.35** | GPR-NIAM$_P$ | 90.57 | 95.23 | 92.84 |
| GPR-NIAM$_T$ | 73.62 | **74.48** | **73.97** | GPR-NIAM$_T$ | **96.03** | 96.24 | 96.13 | GPR-NIAM$_T$ | **91.86** | 94.38 | 93.10 |

| (d) Caltech101 | | | (e) DTD | | | (f) FGVCAircraft | | |
|---|---|---|---|---|---|---|---|---|
| **Method** | **Base** | **Novel** | **HM** | **Method** | **Base** | **Novel** | **HM** | **Method** | **Base** | **Novel** | **HM** |
| CLIP | 82.27 | 94.05 | 87.77 | CLIP | 53.35 | 59.05 | 56.06 | CLIP | 26.51 | 31.21 | 28.67 |
| PromptFL | 87.07 | 93.03 | 89.95 | PromptFL | 56.36 | 52.53 | 54.38 | PromptFL | 26.03 | 30.61 | 28.13 |
| RPO | 84.95 | 93.80 | 89.16 | RPO | 53.12 | **60.02** | 56.36 | RPO | 26.51 | 30.37 | 28.31 |
| FedTPG | 85.38 | 93.37 | 89.20 | FedTPG | 53.35 | 50.00 | 51.62 | FedTPG | 21.17 | 26.35 | 23.48 |
| TCP | 88.41 | **94.16** | 91.19 | TCP | 58.10 | 57.60 | 57.85 | TCP | 28.37 | 32.23 | 30.18 |
| PromptFolio | 82.41 | 92.61 | 87.21 | PromptFolio | 53.58 | 55.55 | 54.55 | PromptFolio | 28.13 | **34.15** | **30.85** |
| DP-FPL | 87.57 | 93.54 | 90.46 | DP-FPL | 51.85 | 45.65 | 48.55 | DP-FPL | 26.63 | 29.35 | 27.92 |
| FedDISC | 22.37 | 81.39 | 35.09 | FedDISC | 65.23 | 41.23 | 49.87 | FedDISC | 18.49 | 17.61 | 18.03 |
| FedELMY | 85.59 | 90.40 | 87.93 | FedELMY | 40.27 | 11.95 | 18.43 | FedELMY | 23.45 | 5.46 | 8.86 |
| GPR-NIAM$_P$ | **91.24** | 93.88 | **92.54** | GPR-NIAM$_P$ | **67.59** | 52.89 | 59.34 | GPR-NIAM$_P$ | **29.69** | 30.81 | 30.23 |
| GPR-NIAM$_T$ | 90.60 | 94.05 | 92.30 | GPR-NIAM$_T$ | 65.62 | 57.79 | **61.45** | GPR-NIAM$_T$ | 29.39 | 29.96 | 29.67 |

| (g) Flowers102 | | | (h) StanfordCars | | | (i) UCF101 | | |
|---|---|---|---|---|---|---|---|---|
| **Method** | **Base** | **Novel** | **HM** | **Method** | **Base** | **Novel** | **HM** | **Method** | **Base** | **Novel** | **HM** |
| CLIP | 62.39 | 78.10 | 69.36 | CLIP | 56.87 | 69.40 | 62.51 | CLIP | 68.12 | 71.92 | 69.97 |
| PromptFL | 72.88 | 75.78 | 74.30 | PromptFL | 57.31 | 67.58 | 62.02 | PromptFL | 69.55 | 68.90 | 69.22 |
| RPO | 72.01 | 76.63 | 74.24 | RPO | 57.64 | 66.73 | 61.85 | RPO | 67.01 | 71.45 | 69.16 |
| FedTPG | 70.84 | 77.26 | 73.91 | FedTPG | 56.89 | **69.55** | 62.59 | FedTPG | 68.02 | 65.83 | 66.91 |
| TCP | 73.61 | 77.94 | 75.71 | TCP | 57.79 | 69.28 | 63.01 | TCP | 70.88 | 70.43 | 70.65 |
| PromptFolio | 75.51 | 76.42 | 75.96 | PromptFolio | 58.97 | 69.05 | 63.61 | PromptFolio | 71.29 | 67.84 | 69.52 |
| DP-FPL | 68.97 | 71.43 | 70.08 | DP-FPL | 53.99 | 63.65 | 58.43 | DP-FPL | 73.61 | 69.80 | 71.66 |
| FedDISC | 25.44 | 32.47 | 28.52 | FedDISC | 42.59 | 51.23 | 46.51 | FedDISC | 70.12 | 53.23 | 60.51 |
| FedELMY | **81.63** | 57.05 | 67.16 | FedELMY | 60.26 | 49.94 | 54.62 | FedELMY | **76.72** | 51.90 | 61.92 |
| GPR-NIAM$_P$ | 79.30 | 77.26 | 78.26 | GPR-NIAM$_P$ | **60.81** | 66.98 | 63.75 | GPR-NIAM$_P$ | 75.88 | 68.85 | 72.19 |
| GPR-NIAM$_T$ | 80.17 | **79.15** | **79.65** | GPR-NIAM$_T$ | 60.14 | 68.65 | **64.12** | GPR-NIAM$_T$ | 75.19 | **75.63** | **75.41** |

performance on the eight datasets. Specifically, GPR-NIAM$_P$ and GPR-NIAM$_T$ achieve the top two performances on the base classes, ranking first and second, respectively. In terms of novel class generalization, GPR-NIAM$_T$ outperforms all baselines, demonstrating its superior capability to adapt to unseen categories. Furthermore, GPR-NIAM$_T$ also achieves the highest overall performance in terms of harmonic mean (HM), indicating a strong balance between knowledge retention and generalization. These results highlight the robustness and adaptability of the proposed method under highly heterogeneous federated settings.

### A.2.2 Leave-One-Domain-Out Generalization with Increased Client Diversity

To validate the effectiveness of the proposed GPR-NIAM under more challenging scenarios, we conduct experiments in a leave-one-domain-out generalization setting [] using the Office-Home and DomainNet datasets. In this setting, the data from each domain is randomly partitioned among 5 clients, and for each evaluation, one domain is held out entirely for testing while the remaining domains are used for federated training. As shown in Table 7, GPR-NIAM$_P$ and GPR-NIAM$_T$ consistently outperform existing baselines across both datasets. For the Office-Home, GPR-NIAM$_T$ achieves the highest average accuracy, with GPR-NIAM$_P$ closely following. Notably, GPR-NIAM$_T$ yields the best performance on the "RealWorld" and "Product" domains, while GPR-NIAM$_P$ leads on "Art" and "Clipart". For the DomainNet, both variants of GPR-NIAM also demonstrate strong cross-domain

Table 7: Performance comparison between `GPR-NIAM` and baselines on the Office-Home and DomainNet datasets under the leave-one-domain-out generalization setting, where the data from each domain is randomly partitioned among 5 clients.

| Method | Office-Home | | | | | DomainNet | | | | | | |
|---|---|---|---|---|---|---|---|---|---|---|---|---|
| | Art | Clipart | Product | RealWorld | Average | Clipart | Infograph | Painting | Quickdraw | Real | Sketch | Average |
| CLIP | 74.57 | 62.33 | 80.17 | 79.38 | 74.08 | 92.87 | 77.02 | 85.35 | 43.06 | 98.61 | 94.22 | 81.85 |
| PromptFL | 73.71 | 64.49 | 82.81 | 80.39 | 75.35 | 94.93 | 80.09 | **87.94** | 52.68 | 99.00 | 95.63 | 85.04 |
| RPO | 73.83 | 62.13 | 79.95 | 79.36 | 73.81 | 93.66 | 77.31 | 84.65 | 44.46 | 98.83 | 93.87 | 82.13 |
| FedTPG | 74.90 | 65.54 | 82.85 | 80.37 | 75.91 | 94.35 | 79.76 | 87.40 | 48.05 | 98.92 | **95.89** | 84.06 |
| TCP | 75.77 | 65.10 | 83.10 | 80.99 | 76.24 | 93.92 | 79.72 | 87.67 | 46.82 | 98.96 | 95.00 | 83.68 |
| PromptFolio | 75.81 | 65.36 | 83.53 | 80.39 | 76.27 | **95.51** | **81.00** | 89.33 | 48.82 | 98.99 | 95.20 | 84.80 |
| DP-FPL | 74.82 | 63.55 | 82.85 | 78.79 | 75.00 | 94.77 | 80.20 | 89.00 | 48.35 | 98.78 | 94.82 | 84.32 |
| FedDISC | 71.19 | 62.48 | 81.31 | 78.35 | 73.48 | 94.23 | 80.25 | 83.85 | 44.21 | 98.39 | 92.49 | 82.23 |
| FedELMY | 75.45 | 64.29 | 82.36 | 77.58 | 74.92 | 94.15 | 75.48 | 86.89 | 47.39 | 98.72 | 95.13 | 82.96 |
| `GPR-NIAM`$_P$ | **78.16** | **72.76** | 89.09 | 83.68 | 80.92 | 94.45 | 80.60 | 87.50 | **58.68** | **99.10** | 94.91 | 85.87 |
| `GPR-NIAM`$_T$ | 77.66 | 71.95 | **89.47** | **85.93** | **81.25** | 95.19 | 80.49 | 87.23 | 57.96 | 99.07 | 95.37 | **85.88** |

generalization, achieving top results in multiple domains. These results highlight the robustness of our method in handling both domain shift and non-IID data across clients. In summary, the proposed `GPR-NIAM` exhibits superior generalization performance under leave-one-domain-out conditions, demonstrating its effectiveness in federated scenarios with high domain and client-level variability.

### A.2.3 Performance Comparison across Different Local Training Epoch Settings

Table 8: Performance comparison between `GPR-NIAM` and baselines on StanfordCars and UCF101 with local training epochs $\in \{1, 5, 20\}$ in the base-to-base/novel generalization setting.

| (g) StanfordCars (Epochs=1) | | | | (h) StanfordCars (Epochs=5) | | | | (i) StanfordCars (Epochs=20) | | | |
|---|---|---|---|---|---|---|---|---|---|---|---|
| Method | Base | Novel | HM | Method | Base | Novel | HM | Method | Base | Novel | HM |
| CLIP | 56.87 | 69.40 | 62.51 | CLIP | 56.87 | 69.40 | 62.51 | CLIP | 56.87 | 69.40 | 62.51 |
| PromptFL | 58.87 | 68.40 | 63.27 | PromptFL | 60.39 | 67.35 | 63.68 | PromptFL | 59.74 | 67.83 | 63.52 |
| RPO | 56.39 | 69.03 | 62.07 | RPO | 57.12 | 69.70 | 62.79 | RPO | 57.42 | **70.06** | 63.11 |
| FedTPG | 57.09 | 69.73 | 62.78 | FedTPG | 58.14 | 69.65 | 63.38 | FedTPG | 58.69 | 69.25 | 63.54 |
| TCP | 58.04 | 69.40 | 63.22 | TCP | 58.72 | 69.91 | 63.82 | TCP | 59.29 | 68.68 | 63.64 |
| PromptFolio | 58.69 | **69.83** | 63.78 | PromptFolio | 60.04 | 69.78 | 64.54 | PromptFolio | 61.46 | 69.10 | 65.06 |
| DP-FPL | 49.30 | 62.85 | 55.25 | DP-FPL | 59.02 | 68.33 | 63.33 | DP-FPL | 59.24 | 64.35 | 61.69 |
| `GPR-NIAM`$_P$ | 60.61 | 67.58 | 63.91 | `GPR-NIAM`$_P$ | 62.09 | 67.05 | 64.48 | `GPR-NIAM`$_P$ | **64.46** | 66.38 | 65.41 |
| `GPR-NIAM`$_T$ | **61.54** | 67.40 | 64.33 | `GPR-NIAM`$_T$ | 60.99 | 69.45 | **64.94** | `GPR-NIAM`$_T$ | 63.01 | 69.00 | **65.87** |
| (j) UCF101 (Epochs=1) | | | | (k) UCF101 (Epochs=5) | | | | (l) UCF101 (Epochs=20) | | | |
| Method | Base | Novel | HM | Method | Base | Novel | HM | Method | Base | Novel | HM |
| CLIP | 68.12 | 71.92 | 69.97 | CLIP | 68.12 | 71.92 | 69.97 | CLIP | 68.12 | 71.92 | 69.97 |
| PromptFL | 69.55 | 69.01 | 69.28 | PromptFL | 73.03 | 72.08 | 72.55 | PromptFL | 68.07 | 64.24 | 66.10 |
| RPO | 65.96 | 70.55 | 68.17 | RPO | 66.80 | 71.50 | 69.07 | RPO | 68.75 | 71.13 | 69.92 |
| FedTPG | 64.69 | 64.08 | 64.39 | FedTPG | 68.86 | 67.47 | 68.16 | FedTPG | 68.86 | 66.73 | 67.77 |
| TCP | 73.82 | 72.47 | 73.13 | TCP | 75.83 | 72.27 | 74.00 | TCP | 76.09 | 73.00 | 74.51 |
| PromptFolio | 73.72 | 69.80 | 71.71 | PromptFolio | 76.56 | 70.12 | 73.20 | PromptFolio | 78.15 | 68.37 | 72.94 |
| DP-FPL | 72.13 | 70.65 | 71.38 | DP-FPL | 74.08 | 71.76 | 72.91 | DP-FPL | 73.24 | 68.16 | 70.61 |
| `GPR-NIAM`$_P$ | 76.88 | 68.53 | 72.47 | `GPR-NIAM`$_P$ | **79.10** | 68.75 | 73.56 | `GPR-NIAM`$_P$ | **78.89** | 69.38 | 73.83 |
| `GPR-NIAM`$_T$ | **77.78** | **74.62** | **76.17** | `GPR-NIAM`$_T$ | 78.15 | **75.79** | **76.95** | `GPR-NIAM`$_T$ | 78.04 | **76.58** | **77.31** |

To evaluate the effectiveness of the proposed `GPR-NIAM` under different levels of local training effort, this section investigates its performance across varying local training epochs. Specifically, we consider three settings with local training epochs $\in \{1, 5, 20\}$, under a data heterogeneity level ($\beta = 0.5$) and with 10 clients participating in training. As shown in Table 4 and Table 8, both `GPR-NIAM`$_P$ and `GPR-NIAM`$_T$ consistently outperform the baselines across all datasets and training epochs. Notably, `GPR-NIAM`$T$ delivers the best overall performance in terms of harmonic mean (HM) in most settings, which demonstrates superior generalization across both base and novel classes. On Flowers102 and UCF101, `GPR-NIAM`$T$ shows strong robustness even when the number of local epochs increases. This highlights its ability to prevent overfitting to local distributions. Across datasets, both variants of `GPR-NIAM` remain stable and effective even as the local training dynamics change, outperforming methods that are more sensitive to overfitting or class imbalance under prolonged local updates.

### A.2.4 Sensitivity Analysis Regarding the Number of Prompt Tokens.

Table 9: Comparison of different methods under two settings (5×512 and 20×512) on Caltech101, DTD, and UCF101 datasets. Bold numbers indicate the best performance.

| | Caltech101 | | | DTD | | | UCF101 | | |
|---|---|---|---|---|---|---|---|---|---|
| | Base | Novel | HM | Base | Novel | HM | Base | Novel | HM |
| **5×512** | | | | | | | | | |
| PromptFL | 88.41 | **93.97** | 91.11 | 59.25 | 52.77 | 55.83 | 76.78 | 69.91 | 73.18 |
| FedTPG | 86.65 | 93.37 | 89.88 | 60.18 | 56.15 | 58.10 | 74.30 | 70.63 | 72.41 |
| PromptFolio | 89.47 | 93.46 | 91.42 | **61.02** | **56.28** | 57.62 | 74.40 | **70.65** | 72.48 |
| GRP-NIAM_P | **89.90** | 93.37 | **91.60** | 60.96 | 55.95 | **59.12** | **77.98** | 70.39 | **73.99** |
| **20×512** | | | | | | | | | |
| PromptFL | 88.91 | **94.31** | 91.53 | **62.50** | 52.53 | 57.08 | 76.83 | 69.12 | 72.77 |
| FedTPG | 88.84 | 92.86 | 90.81 | 58.10 | 54.83 | 56.41 | 76.30 | 73.41 | 74.41 |
| PromptFolio | 89.97 | 94.22 | 92.05 | 56.13 | 53.98 | 55.03 | 75.72 | 74.41 | 75.06 |
| GRP-NIAM_P | **90.02** | 94.28 | **92.10** | 61.43 | **56.23** | **58.71** | **77.23** | **74.58** | **75.88** |

We have conducted additional sensitivity experiments regarding the number of prompt tokens. Specifically, we adjusted the prompt length to 5×512 and 20×512. The results show that GPR-NIAM remains robust across different prompt lengths, while consistently maintaining an advantage over all baselines in all case.

### A.2.5 Evaluating Method Scalability with Varying Client Numbers

Table 10: Performance comparison between GPR-NIAM and baselines on DTD and Flowers102 datasets under varying number of clients $K \in \{20, 30\}$ in the base-to-base/novel generalization.

| (a) DTD (K=20) | | | | (b) DTD (K=30) | | | |
|---|---|---|---|---|---|---|---|
| **Method** | **Base** | **Novel** | **HM** | **Method** | **Base** | **Novel** | **HM** |
| CLIP | 53.35 | 59.05 | 56.06 | CLIP | 53.35 | 59.05 | 56.06 |
| PromptFL | 54.16 | 54.22 | 54.19 | PromptFL | 52.89 | 56.52 | 54.64 |
| RPO | 53.24 | **60.02** | 56.42 | RPO | 52.77 | **59.66** | 56.00 |
| FedTPG | 57.83 | 54.93 | 56.34 | FedTPG | 52.19 | 58.09 | 54.98 |
| TCP | 58.21 | 55.93 | 57.04 | TCP | 56.82 | 58.42 | 57.60 |
| PromptFolio | 58.18 | 52.91 | 55.41 | PromptFolio | 55.43 | 52.41 | 53.88 |
| DP-FPL | 56.13 | 50.72 | 53.29 | DP-FPL | 57.06 | 51.20 | 53.97 |
| GPR-NIAM$_P$ | **63.31** | 52.29 | 57.27 | GPR-NIAM$_P$ | **60.64** | 51.93 | 55.95 |
| GPR-NIAM$_T$ | 62.38 | 53.86 | **57.81** | GPR-NIAM$_T$ | 60.41 | 56.52 | **58.40** |

| (d) Flowers102 (K=20) | | | | (e) Flowers102 (K=30) | | | |
|---|---|---|---|---|---|---|---|
| **Method** | **Base** | **Novel** | **HM** | **Method** | **Base** | **Novel** | **HM** |
| CLIP | 62.39 | 78.10 | 69.36 | CLIP | 62.39 | 78.10 | 69.36 |
| PromptFL | 74.34 | 74.10 | 74.22 | PromptFL | 72.01 | 74.73 | 73.34 |
| RPO | 71.72 | 76.84 | 74.19 | RPO | 71.13 | 76.42 | 73.68 |
| FedTPG | 74.63 | **78.31** | 76.43 | FedTPG | 74.92 | 78.31 | 76.58 |
| TCP | 75.21 | 77.68 | 76.43 | TCP | 73.46 | 78.73 | 76.01 |
| PromptFolio | 75.51 | **78.31** | 76.88 | PromptFolio | 73.76 | 78.31 | 75.97 |
| DP-FPL | 74.34 | 74.10 | 74.22 | DP-FPL | 69.09 | 75.57 | 72.19 |
| GPR-NIAM$_P$ | 82.79 | 77.05 | 79.82 | GPR-NIAM$_P$ | 81.63 | 78.31 | 79.93 |
| GPR-NIAM$_T$ | **83.96** | 77.05 | **80.36** | GPR-NIAM$_T$ | **82.50** | **78.94** | **80.68** |

This section aims to evaluate the scalability of the proposed GPR-NIAM with respect to the number of participating clients. We consider two settings with $K \in \{20, 30\}$ under a heterogeneity level $\beta = 0.5$ and fixed local training epochs of 5. As shown in Table 10 and Table 11, both GPR-NIAM$_P$ and GPR-NIAM$_T$ consistently outperform all baselines across datasets and client configurations. GPR-NIAM$_T$ achieves the best overall performance on most datasets and demonstrates strong generalization to novel classes. For example, on Flowers102 and UCF101, it achieves the highest harmonic mean (HM) under both client settings. Meanwhile, GPR-NIAM$_P$ maintains strong

Table 11: Performance comparison between GPR-NIAM and baselines on StanfordCars and UCF101 datasets under varying number of clients $K \in \{20, 30\}$ in the base-to-base/novel generalization.

| (g) StanfordCars (K=20) | | | | (h) StanfordCars (K=30) | | | |
|---|---|---|---|---|---|---|---|
| **Method** | **Base** | **Novel** | **HM** | **Method** | **Base** | **Novel** | **HM** |
| CLIP | 56.87 | 69.40 | 62.51 | CLIP | 56.87 | 69.40 | 62.51 |
| PromptFL | 59.82 | 67.03 | 63.22 | PromptFL | 54.42 | 66.33 | 59.79 |
| RPO | 57.22 | **69.65** | 62.83 | RPO | 57.24 | **69.60** | 62.82 |
| FedTPG | 59.24 | 66.73 | 62.76 | FedTPG | 59.42 | 69.08 | 63.88 |
| TCP | 58.77 | 67.78 | 62.95 | TCP | 59.32 | 68.21 | 63.45 |
| PromptFolio | 60.91 | 65.78 | 63.25 | PromptFolio | 59.59 | 65.25 | 62.29 |
| DP-FPL | 57.74 | 65.65 | 61.44 | DP-FPL | 58.37 | 67.75 | 62.71 |
| GPR-NIAM$_P$ | 60.04 | 68.23 | 63.87 | GPR-NIAM$_P$ | 60.09 | 67.40 | 63.54 |
| GPR-NIAM$_T$ | **61.02** | 69.40 | **64.94** | GPR-NIAM$_T$ | **60.14** | 69.38 | **64.43** |
| (j) UCF101 (K=20) | | | | (k) UCF101 (K=30) | | | |
| **Method** | **Base** | **Novel** | **HM** | **Method** | **Base** | **Novel** | **HM** |
| CLIP | 68.12 | 71.92 | 69.97 | CLIP | 68.12 | 71.92 | 69.97 |
| PromptFL | 75.67 | 68.75 | 72.04 | PromptFL | 71.34 | 66.04 | 68.59 |
| RPO | 66.80 | 71.34 | 69.00 | RPO | 66.75 | 71.39 | 68.99 |
| FedTPG | 74.88 | 74.78 | 74.83 | FedTPG | 76.09 | 74.62 | 75.35 |
| TCP | 74.08 | 73.83 | 73.96 | TCP | 73.50 | 75.58 | 74.53 |
| PromptFolio | 74.45 | 71.76 | 73.08 | PromptFolio | 77.09 | 74.04 | 75.54 |
| DP-FPL | 75.51 | 69.75 | 72.52 | DP-FPL | 72.71 | 68.96 | 70.79 |
| GPR-NIAM$_P$ | 76.41 | 69.01 | 72.52 | GPR-NIAM$_P$ | 76.62 | 72.50 | 74.50 |
| GPR-NIAM$_T$ | **77.04** | **76.21** | **76.62** | GPR-NIAM$_T$ | **77.62** | **76.05** | **76.83** |

performance on base classes. These results show that GPR-NIAM maintains high performance and stability even as the number of clients increases, demonstrating its scalability and effectiveness in more challenging cases.

### A.2.6 Evaluation of Module Contributions

Table 12: Ablation study on the effectiveness of NIAM and CSCR modules under the base-to-base/novel generalization. Results are reported on four datasets with PromptFL as the baseline.

| (a) DTD | | | | (b) Flowers102 | | | |
|---|---|---|---|---|---|---|---|
| **Method** | **Base** | **Novel** | **HM** | **Method** | **Base** | **Novel** | **HM** |
| PromptFL | 55.55 | 56.76 | 56.15 | PromptFL | 78.42 | 76.00 | 77.19 |
| +NIAM | 55.78 | 54.73 | 55.25 | +NIAM | 76.13 | 77.78 | 76.94 |
| +NIAM+CSCR | 68.51 | 52.65 | 59.54 | +NIAM+CSCR | 84.83 | 78.31 | 81.44 |
| (c) StanfordCars | | | | (d) UCF101 | | | |
| **Method** | **Base** | **Novel** | **HM** | **Method** | **Base** | **Novel** | **HM** |
| PromptFL | 60.39 | 67.35 | 63.68 | PromptFL | 73.03 | 72.08 | 72.55 |
| +NIAM | 55.17 | 68.13 | 60.96 | +NIAM | 67.96 | 73.02 | 70.39 |
| +NIAM+CSCR | 62.09 | 67.05 | 64.48 | +NIAM+CSCR | 79.10 | 68.75 | 73.56 |

We evaluate the contributions of the NIAM and CSCR modules by gradually adding them to the PromptFL baseline. Experiments are conducted on four datasets using 10 clients, with heterogeneity level $\beta = 0.5$ and 5 local training epochs. As shown in Table 12, NIAM improves performance on novel classes by preserving generalized knowledge. CSCR enhances base class performance through visual supervision. Together, they deliver the best overall performance in terms of harmonic mean (HM), highlighting their complementary effects.

### A.2.7 Sensitive Analysis of Textual Prompt Template

The model's performance is not highly sensitive to the choice of textual prompt templates. Following prior works such as CLIP and PromptFL, we use "a photo of a [CLASS]" as the default template. Since our method mainly optimizes the learnable prompt tokens ($T_P$) while keeping the text tokens ($T_T$) fixed, the initial template primarily serves as a starting point for semantic alignment and has limited impact on the final performance. We also conducted a small-scale experiment comparing different templates (e.g., "a picture of a [CLASS]", "an image showing [CLASS]"), and the variation in Top-1 accuracy was less than 1%, indicating that the **GPR-NIAM** framework is robust to the choice of templates.

Table 13: Prompt template comparison on Caltech101, DTD, and UCF101. Best numbers in **bold**.

| Template | Caltech101 | | | DTD | | | UCF101 | | |
|---|---|---|---|---|---|---|---|---|---|
| | Base | Novel | HM | Base | Novel | HM | Base | Novel | HM |
| A photo of a [CLASS] | 91.59 | 94.22 | **92.89** | 68.40 | 53.50 | 60.04 | 78.57 | **70.28** | 74.19 |
| A picture of a [CLASS] | 91.43 | **94.34** | 92.86 | 67.37 | 52.83 | 59.22 | 78.61 | 69.54 | 73.79 |
| An image showing [CLASS] | **91.52** | 93.88 | 92.68 | **68.57** | **54.12** | **60.49** | **78.77** | 70.14 | **74.20** |

