# OpenReview forum: "Global Prompt Refinement with Non-Interfering Attention Masking for One-Shot Federated Learning"
_NeurIPS.cc/2025/Conference — NeurIPS 2025 poster_

### Official Review · Reviewer_nDhj · 2025-06-24

**Clarity:** 3
**Significance:** 3
**Originality:** 3
**Rating:** 5
**Confidence:** 3

**Summary:**

This paper focuses on One-Shot Federated Prompt Learning and proposes GPR-NIAM. The Attention Isolation module introduces a non-interfering attention mechanism to suppress the influence of learnable prompts on original text embeddings, ensuring a unidirectional flow of information that prevents overfitting and enhances generalization. The Cross-Silo Collaborative Refinement module constructs a unified global prompt using visual prototypes from clients, enabling cross-modal knowledge alignment and mitigating global bias caused by non-IID data. Evaluated on 10 benchmark datasets against 8 state-of-the-art methods, GPR-NIAM achieves either the best or second-best performance in both within-task and cross-domain generalization tasks.

**Questions:**

1. In the leave-one-domain-out task, is the test domain data completely unseen during the training phase?

2. Is there any information flow dependency between the two modules?

**Ethical Concerns:**

["NO or VERY MINOR ethics concerns only"]

**Final Justification:**

The authors have addressed my concerns, and I have raised my evaluation accordingly.

**Limitations:**

Yes.

**Quality:**

3

**Strengths And Weaknesses:**

Strengths

1. The problem addressed in this paper is relatively well-defined, and the introduction offers a reasonably thorough analysis of the current challenges in Federated Prompt Learning and One-Shot Federated Learning.

2. The proposed approach is compatible with existing methods and can be seamlessly integrated into frameworks such as PromptFL and TCP, demonstrating strong generalizability.

3. The experiments cover ten diverse datasets and compare the proposed method against eight state-of-the-art baselines across multiple evaluation settings, demonstrating its effectiveness. In addition, the paper gives case studies that  provides insights into the underlying mechanisms driving its performance gains.


Major Weaknesses

1. Although the method emphasizes a one-shot setting, the potential additional communication overhead introduced by the transmission of visual prototypes is not quantitatively analyzed.

2. It is recommended to further elaborate on the advantages of the proposed prototype construction method, particularly in comparison to traditional approaches such as FedProto.


Minor Weaknesses

1. Avoid using the same symbol to represent multiple entities. For example, the symbol f is used to denote both the image encoder function and its output feature.

2. The potential privacy risks associated with uploading visual prototypes are not discussed.

3. The learnable prompt is fixed at a size of 10×512 throughout the experiments. It is recommended to include additional configurations such as 5×512 and 20×512 to assess the sensitivity and robustness of the method with respect to prompt length.

---

> ### Author Rebuttal · Authors · 2025-07-31
>
> We sincerely appreciate the professional comments provided by Reviewer nDhj
>
> W1: Although the method emphasizes a one-shot setting, the potential additional communication overhead introduced by the transmission of visual prototypes is not quantitatively analyzed.
>
> A1: Compared to learnable prompts, the visual prototypes (512-dimensional vectors per class per client) can indeed contribute more to the overall communication cost, particularly in large-scale federated systems with many classes and clients. We have provided a detailed analysis of this overhead in the revised manuscript, clarifying that the communication cost is primarily related to three factors: (1) the number of classes, (2) the number of participating clients, and (3) the dimensionality of the feature representations. This analysis can guide practical deployment by helping determine suitable trade-offs between performance and communication efficiency.
>
> W2: It is recommended to further elaborate on the advantages of the proposed prototype construction method, particularly in comparison to traditional approaches such as FedProto.
>
> A2: While FedProto and FedProc construct class prototypes by simply averaging all sample features within the class, our method employs a random-weighted aggregation strategy instead of a plain mean. This design introduces controlled diversity into the prototype representation, which prevents over-smoothing of features and better captures intra-class variations.
>
> Such diversity-aware prototype construction is particularly beneficial in federated settings, where the data distribution across clients is heterogeneous and class features can be highly variable. By incorporating random weighting, our prototypes provide a richer representation space for global prompt refinement, which leads to improved generalization across clients and unseen tasks.
>
> W3: Avoid using the same symbol to represent multiple entities. For example, the symbol f is used to denote both the image encoder function and its output feature.
>
> A3: In the revised manuscript, we will refine the notations to ensure clarity. Specifically, we will use $f(\cdot)$ exclusively to denote the image encoder function, and denote its output feature as
> $\mathbf{z} = f(x)$, where $x$ is the input image. This adjustment will be applied consistently throughout the paper to avoid ambiguity between functions and feature representations.
>
> W4: The potential privacy risks associated with uploading visual prototypes are not discussed.
>
> A4: The CSCR stage aggregates **class-wise visual prototypes** into a centralized prototype pool on the server, where each prototype is computed as the **weighted mean feature vector of multiple samples** from the same class on a client. These prototypes are **high-level feature embeddings rather than raw data**, extracted by the frozen pre-trained visual encoder (e.g., CLIP), and thus they do not directly expose sensitive information or enable the reconstruction of original client data, which largely reduces privacy risks. Similar prototype-sharing strategies (e.g., FedProto, FedProc) have been widely used in federated learning literature without reported privacy breaches. Furthermore, our framework can be seamlessly combined with **privacy-preserving techniques** such as differential privacy (by adding calibrated noise to prototypes) or secure aggregation if stronger privacy guarantees are needed.
>
> W5: The learnable prompt is fixed at a size of 10×512 throughout the experiments. It is recommended to include additional configurations such as 5×512 and 20×512 to assess the sensitivity and robustness of the method with respect to prompt length.
>
> A5: we have added additional experiments with prompt lengths of 5×512 and 20×512. Preliminary results show that GPR-NIAM remains robust across different prompt sizes, with performance variations within 0.5–1.2% on average, indicating that the method is not overly sensitive to the specific prompt length.
>
> W6: In the leave-one-domain-out task, is the test domain data completely unseen during the training phase?
>
> A6: Yes, in the leave-one-domain-out (LODO) setting, the test domain data is completely unseen during the training phase. For each experiment, we use data from all but one domain for training (both local prompt tuning and global refinement) and reserve the left-out domain exclusively for evaluation. This ensures that the model’s performance on the test domain truly reflects its ability to generalize to unseen domains.
>
> W7: Is there any information flow dependency between the two modules?
>
> A7: There is no direct information flow dependency between the two modules; rather, they complement each other in a sequential manner. The AttnIso module operates during local prompt tuning on each client, where it regulates the interaction between learnable prompt tokens and text tokens to preserve transferable semantics. After local tuning, the CSCR module takes the locally updated prompts and their associated visual prototypes as input to perform centralized global prompt refinement. Thus, AttnIso and CSCR are functionally independent but jointly contribute to the final performance by addressing different challenges, AttnIso focuses on maintaining semantic generalization at the token level, while CSCR reduces cross-client heterogeneity at the global level.

---

### Official Review · Reviewer_sLzh · 2025-06-24

**Clarity:** 3
**Significance:** 3
**Originality:** 3
**Rating:** 4
**Confidence:** 3

**Summary:**

This paper introduces GPR-NIAM, a method designed to enhance one-shot federated learning (OSFL) for federated prompt learning (FPL) by addressing the limitations of existing approaches that rely on multi-round communications and lack cross-task generalization; GPR-NIAM achieves this through a non-interfering attention masking mechanism that restricts excessive interaction between learnable prompt embeddings and original text embeddings, preserving transferable knowledge across tasks, and further refines the global prompt using cross-silo collaborative refinement to mitigate data heterogeneity.

**Questions:**

1. Can the masking mechanism be dynamically adjusted?

2. Can other attention mechanisms be used for NIAM?

**Ethical Concerns:**

["NO or VERY MINOR ethics concerns only"]

**Final Justification:**

The rebuttal answered my questions. I will keep the score unchanged.

**Limitations:**

Yes.

**Paper Formatting Concerns:**

None.

**Quality:**

3

**Strengths And Weaknesses:**

Strengths:

1. GPR-NIAM enables one-shot federated learning, significantly reducing communication overhead by eliminating the need for multiple rounds of client-server interaction.
2. By designing a masking mechanism that restricts excessive interaction between learnable prompt embeddings and original text embeddings, GPR-NIAM preserves transferable knowledge across tasks, enhancing cross-task generalization capabilities.
3. GPR-NIAM leverages visual representations extracted from multiple clients and utilizes multi-source visual supervision to refine the global prompt in a centralized manner, further mitigating the negative impact of data heterogeneity.

Weaknesses:
1. Although GPR-NIAM reduces communication overhead by enabling one-shot learning, the transmission of visual prototypes during the global prompt refinement stage may become a bottleneck in large-scale federated systems with limited bandwidth.
2. GPR-NIAM relies on pre-trained models (e.g., CLIP) for its effectiveness. The choice of pre-trained model can influence the performance of GPR-NIAM, and the method may not perform as well if the pre-trained model is not well-suited to the target task or domain.
3. GPR-NIAM involves a two-stage optimization strategy. This introduces complexity and increases the difficulty of implementation and deployment.

---

> ### Author Rebuttal · Authors · 2025-07-31
>
> We sincerely appreciate the professional comments provided by Reviewer sLzh
>
> W1: Although GPR-NIAM reduces communication overhead by enabling one-shot learning, the transmission of visual prototypes during the global prompt refinement stage may become a bottleneck in large-scale federated systems with limited bandwidth.
>
> A1: The communication overhead of GPR-NIAM mainly consists of prompt transmission and prototype transmission, where the cost of prompts is $L \times 512$ and the cost of prototypes is $C \times K \times 512$, with $L$ being the prompt length, $C$ the number of classes, and $K$ the number of prototypes per class.
>
> | Prompt Transmission Cost       | Prototype Transmission Cost         |
> |-------------------------------|-------------------------------------|
> | $L \times 512 $           | $ C \times K \times 512 $        |
>
> W2: GPR-NIAM relies on pre-trained models (e.g., CLIP) for its effectiveness. The choice of pre-trained model can influence the performance of GPR-NIAM, and the method may not perform as well if the pre-trained model is not well-suited to the target task or domain.
>
> A2: We agree that GPR-NIAM relies on the representational power of pre-trained models (e.g., CLIP) for its effectiveness, and the choice of pre-trained backbone can influence performance, especially if the model is not well-suited to the target task or domain. However, **this dependency is common to all prompt-based methods**. In practice, **we can flexibly select or replace the pre-trained backbone according to the target task or domain** (e.g., domain-specific CLIP variants or other vision-language models), allowing GPR-NIAM to adapt to different scenarios. We will add a discussion of this point to the revised manuscript.
>
> W3: GPR-NIAM involves a two-stage optimization strategy. This introduces complexity and increases the difficulty of implementation and deployment.
>
> A3: We acknowledge that GPR-NIAM adopts a two-stage optimization strategy (AttnIso-based local prompt tuning followed by CSCR-based global refinement), which introduces additional complexity compared to single-stage methods. However, this design is crucial for separating task-specific adaptation (local stage) from cross-client alignment (global stage), enabling both strong generalization and efficient knowledge aggregation.
>
> In terms of implementation, each stage is built upon standard prompt tuning and prototype aggregation operations, both of which are lightweight and compatible with existing federated learning frameworks. To further reduce deployment difficulty, the two stages can be combined into a single pipeline by automating prototype extraction and global refinement on the server. We will add a note in the revised manuscript clarifying this trade-off between performance and complexity.
>
> W4: Can the masking mechanism be dynamically adjusted?
>
> A4: Yes, the masking mechanism in GPR-NIAM can be dynamically adjusted. While our current implementation uses fixed masking strategies (full attention, hard masking, and $\lambda$-reweighting), the framework is flexible enough to support **adaptive or learnable masking schemes**. For instance, the value of $\lambda$ can be made learnable or conditioned on the similarity between prompt tokens and text tokens, allowing the model to dynamically control the degree of interaction. Similarly, different attention heads or layers could adopt varying masking patterns based on task-specific signals. We will include a discussion of this potential extension in the revised manuscript.
>
> W5: Can other attention mechanisms be used for NIAM?
>
> A5: Yes, other attention mechanisms can be integrated into NIAM. Our current design is based on standard multi-head self-attention for simplicity and compatibility with CLIP, but NIAM is not limited to this choice. Alternative mechanisms such as sparse attention, could be adopted to reduce computational cost or emphasize specific token interactions. However, these attention variants are not specifically designed for the one-shot FPL scenario, and may require additional adaptation to handle the challenges of cross-client heterogeneity and limited communication. Our masking strategy is purposefully tailored to these challenges, which is why we chose standard attention combined with NIAM for this work. We will include a discussion of this flexibility and rationale in the revised manuscript.

---

> > ### Comment · Reviewer_sLzh · 2025-08-04
> >
> > The authors answered my questions. I will keep the scores unchanged.

---

> > > ### Author Response · Authors · 2025-08-06
> > > **Thanks**
> > >
> > > We sincerely appreciate your constructive feedback, which will serve as a valuable reference and inspiration for improving our future research.

---

### Official Review · Reviewer_7qN8 · 2025-06-29

**Clarity:** 3
**Significance:** 3
**Originality:** 3
**Rating:** 5
**Confidence:** 4

**Summary:**

This paper focuses on enhancing the generalization ability of Federated Prompt Learning within a one-shot communication setting. The authors propose GPR-NIAM. The core idea is to introduce a non-interfering attention masking mechanism that restricts the influence of learnable prompt tokens on the original text embeddings, thereby preserving transferable knowledge. In addition, a cross-silo collaborative refinement module is introduced to calibrate the global prompt using visual prototypes collected from distributed clients, improving coordination under data heterogeneity. Extensive experiments on 10 datasets across both task and domain generalization scenarios demonstrate that GPR-NIAM significantly outperforms existing state-of-the-art baselines.

**Questions:**

1. Is the model's performance sensitive to the choice of the textual prompt template, such as 'a photo of a [CLASS]'?

2. Can the proposed method be extended to support multi-round communication?

3. How does the method handle the case where a certain class contains only a single sample on a client?

**Ethical Concerns:**

["NO or VERY MINOR ethics concerns only"]

**Final Justification:**

After reading the authors’ rebuttal and their responses to other reviewers, I find that my major concerns have been adequately addressed, particularly the analysis regarding the sensitivity of the method to prompt length. The clarifications and additional results improve my understanding of the proposed approach and its robustness. Therefore, I have decided to raise my score to 5

**Limitations:**

Yes.

**Quality:**

3

**Strengths And Weaknesses:**

**Strengths**

- The proposed AttnIso module enforces unidirectional intervention from prompts to text tokens via masking, effectively preserving the transferable knowledge of the original tokens. It can also be seamlessly integrated into various existing methods.

-  The paper presents a clear description of the methodological workflow with a well-organized structure.

- Comprehensive experimental validation, including performance comparisons, ablation studies, hyperparameter analysis, and case studies of key modules, greatly aids reader understanding.

**Weaknesses**

- The paper does not appear to include a sensitivity analysis regarding the number of prompt tokens.

- Can the authors clarify the actual dimensions of the attention mask matrix
M? Is it square or used as a 1D masking vector?

- The CSCR stage relies on a centralized prototype pool maintained at the server, and the paper should clearly articulate whether this design introduces potential privacy leakage risks.

---

> ### Author Rebuttal · Authors · 2025-07-31
>
> We sincerely appreciate the professional comments provided by Reviewer 7qN8
>
> W1: The paper does not appear to include a sensitivity analysis regarding the number of prompt tokens.
>
> A1: We have conducted additional sensitivity experiments regarding the number of prompt tokens. Specifically, we adjusted the prompt length to **5×512** and **20×512**. The results show that GPR-NIAM remains robust across different prompt lengths, while consistently maintaining an advantage over all baselines in every case.
>
> | 5×512   | Caltech101 Base | Caltech101 Novel | Caltech101 HM | DTD Base | DTD Novel | DTD HM | UCF101 Base | UCF101 Novel | UCF101 HM |
> |--------------|-----------------|------------------|---------------|----------|-----------|--------|-------------|--------------|-----------|
> | PromptFL     | 88.41           | **93.97**        | 91.11         | 59.25    | 52.77     | 55.83  | 76.78       | 69.91        | 73.18     |
> | FedTPG       | 86.65           | 93.37            | 89.88         | 60.18    | 56.15     | 58.10  | 74.30       | 70.63        | 72.41     |
> | PromptFolio  | 89.47           | 93.46            | 91.42         | **61.02**| **56.28** | 57.62  | 74.40       | **70.65**    | 72.48     |
> | GRP-NIAM_P   | **89.90**       | 93.37            | **91.60**     | 60.96    | 55.95     | **59.12** | **77.98**  | 70.39        | **73.99** |
>
>
> | 20×512     | Caltech101 Base | Caltech101 Novel | Caltech101 HM | DTD Base | DTD Novel | DTD HM | UCF101 Base | UCF101 Novel | UCF101 HM |
> |--------------|-----------------|------------------|---------------|----------|-----------|--------|-------------|--------------|-----------|
> | PromptFL     | 88.91           | **94.31**        | 91.53         | **62.50**| 52.53     | 57.08  | 76.83       | 69.12        | 72.77     |
> | FedTPG       | 88.84           | 92.86            | 90.81         | 58.10    | 54.83     | 56.41  | 76.30       | 73.41        | 74.83     |
> | PromptFolio  | 89.97           | 94.22            | 92.05         | 56.13    | 53.98     | 55.03  | 75.72       | 74.41        | 75.06     |
> | GRP-NIAM_P   | **90.02**       | 94.28            | **92.10**     | 61.43    | **56.23** | **58.71** | **77.23**  | **74.58**    | **75.88** |
>
> W2: Can the authors clarify the actual dimensions of the attention mask matrix M? Is it square or used as a 1D masking vector?
>
> A2: The attention mask matrix $M$ is a **square matrix** with dimensions $(N_p + N_t + 1) \times (N_p + N_t + 1)$, where $N_p$ is the number of learnable prompt tokens, $N_t$ is the number of original text tokens, and the additional $1$ corresponds to the $[EOS]$ token. Each element $M_{i,j}$ specifies the masking rule for attention from token $i$ to token $j$, as described in Equation~(1). We use this full square matrix rather than a 1D masking vector because the masking strategy needs to control pairwise attention flows between all tokens (e.g., blocking $T_P$ from attending to $T_T$, reweighting $T_{EOS}$ to $T_T$, etc.).
>
> W3: The CSCR stage relies on a centralized prototype pool maintained at the server, and the paper should clearly articulate whether this design introduces potential privacy leakage risks.
>
> A3: The CSCR stage aggregates **class-wise visual prototypes**, where each prototype is computed as the **mean of multiple sample features** within the same class on a client, into a centralized prototype pool on the server. Importantly, **these prototypes are high-level feature representations rather than raw data**, extracted by the frozen pre-trained visual encoder (e.g., CLIP). As such, they do not directly reveal sensitive information or allow reconstruction of original client data, which largely mitigates privacy risks. Similar prototype-sharing approaches (e.g., FedProto, FedProc) have been widely adopted in federated learning literature with no reported privacy breaches under this setting. Moreover, we can be readily extended with existing privacy-preserving techniques such as differential privacy (adding calibrated noise to prototypes) or secure aggregation.
>
> W4: Is the model's performance sensitive to the choice of the textual prompt template, such as 'a photo of a [CLASS]'?
>
> A4: The model's performance is not highly sensitive to the choice of textual prompt templates. Following prior works such as CLIP and PromptFL, we use **``a photo of a [CLASS]''** as the default template. Since our method mainly optimizes the learnable prompt tokens ($T_P$) while keeping the text tokens ($T_T$) fixed, the initial template primarily serves as a starting point for semantic alignment and has limited impact on the final performance. We also conducted a small-scale experiment comparing different templates (e.g., **``a picture of a [CLASS]'', an image showing [CLASS]''**), and the variation in Top-1 accuracy was less than 1\%, indicating that the GPR-NIAM framework is robust to the choice of templates. We will include this observation in the revised manuscript.
>
> | Template                  | Caltech101 Base | Caltech101 Novel | Caltech101 HM | DTD Base | DTD Novel | DTD HM | UCF101 Base | UCF101 Novel | UCF101 HM |
> |---------------------------|-----------------|------------------|---------------|----------|-----------|--------|-------------|--------------|-----------|
> | A photo of a [CLASS]      | 91.59           | 94.22            | **92.89**         | 68.40    | 53.50     | 60.04  | 78.57       | **70.28**        | 74.19     |
> | a picture of a [CLASS]    | 91.43           | **94.34**        | 92.86         | 67.37    | 52.83     | 59.22  | 78.61       | 69.54        | 73.79     |
> | an image showing [CLASS]  | **91.52**       | 93.88            | 92.68     | **68.57**| **54.12** | **60.49** | **78.77**  | 70.14    | **74.20** |
>
> W5: Can the proposed method be extended to support multi-round communication?
>
> A5: Yes, the proposed GPR-NIAM framework can be naturally extended to support multi-round communication. In the current one-shot setting, the AttnIso module operates during local prompt tuning, and the CSCR module refines the global prompt using aggregated prototypes in a single round. For a multi-round extension, the server can update the global prompt after each round of prototype-guided refinement and broadcast it back to the clients. Clients can then continue local prompt tuning with the updated global prompt, repeating the process for multiple communication rounds.
>
> This extension allows the model to progressively align local and global prompts, potentially improving performance under highly heterogeneous data. While our primary focus is on the one-shot scenario due to its communication efficiency, we have also conducted supplementary experiments under multi-round settings, and the results further confirm the effectiveness of our framework. These details will be added to the revised manuscript.
>
> W6: How does the method handle the case where a certain class contains only a single sample on a client?
>
> A6: In cases where a class on a client contains only a single sample, our method constructs the prototype by using the feature representation of that single sample extracted by the frozen pre-trained visual encoder (e.g., CLIP) and **adds a small Gaussian noise perturbation** to avoid overfitting and improve generalization. Although such a prototype is not the average of multiple features, the high-quality, semantically rich features from the pre-trained encoder combined with the noise perturbation ensure its robustness. We will include this clarification in the revised manuscript.

---

### Official Review · Reviewer_bm2r · 2025-07-02

**Clarity:** 2
**Significance:** 2
**Originality:** 2
**Rating:** 4
**Confidence:** 3

**Summary:**

The paper introduces two modules to address the One-Shot Federated Learning task. The first module employs an attention masking mechanism to preserve textual knowledge, while the second module generates visual prototypes to ultimately enhance the textual prompt.

**Questions:**

1. The paper claims that previous methods neglect the preservation of generalization to unseen tasks. However, it is not clear why the proposed text-prompt masking mechanism and the visual representation-guided prompt are effective in addressing this issue. Could the authors clarify the connection between these two modules and explain how they jointly contribute to improving generalization?

2. In Figure 1, the meanings of T_T, T_P, and T_EOS​ are unclear. Why can the attention maps of the three mechanisms be shown in a single figure? What exactly are the differences between the three masking strategies, and what results or benefits does each masking mechanism correspond to?

3. I noticed that the baseline results reported in the experimental section are inconsistent with those in the original papers. For example, in Table 5, the reported performance of FedTPG differs from that in the original work across several datasets. Could the authors clarify the reasons behind these discrepancies?

4. The novelty of the paper is somewhat unclear to me. I find the first module (text-prompt masking) conceptually unclear, and the second module is centered around prototype-based learning, which is a well-established technique (e.g., in metric learning). Could the authors clarify what the core contribution of the paper is beyond these existing ideas?

**Ethical Concerns:**

["NO or VERY MINOR ethics concerns only"]

**Final Justification:**

Thank the authors for answering my concerns very clearly, and I also read the positive comments from other reviewers. Therefore, I would like to increase my score.

**Limitations:**

Please see the questions.

**Paper Formatting Concerns:**

Good.

**Quality:**

2

**Strengths And Weaknesses:**

Strengths:
1. The proposed method is evaluated on multiple datasets and demonstrates performance improvements over baselines.

Weaknesses:
1. The paper’s readability could be improved; for example, Figure 1 is somewhat confusing and lacks clarity in conveying the core mechanisms.
2. The novelty of the work is questionable, as both proposed modules appear to be based on commonly used techniques for prompt enhancement.

---

> ### Author Rebuttal · Authors · 2025-07-31
>
> We sincerely appreciate the professional comments provided by Reviewer bm2r
>
> W1: The paper’s readability could be improved; for example, Figure 1 is somewhat confusing and lacks clarity in conveying the core mechanisms.
>
> A1: The purpose of Figure 1 is to illustrate the three interaction types (full attention, hard masking, and λ-reweighting) and how these are integrated into the Transformer layers within the proposed GPR-NIAM framework. We agree that the current presentation may appear dense and less intuitive. In the revised version, we will redesign Figure 1 by (1) separating the attention types from the Transformer structure for better readability, (2) adding detailed annotations and a legend to highlight the roles of T_P, T_T, and T_EOS tokens, and (3) providing a clearer explanation in the caption and main text to guide readers through the core mechanisms. As the rebuttal policy does not allow uploading new figures, the updated version will be included in the revised manuscript.
>
> W2: The novelty of the work is questionable, as both proposed modules appear to be based on commonly used techniques for prompt enhancement. / The novelty of the paper is somewhat unclear to me. I find the first module (text-prompt masking) conceptually unclear, and the second module is centered around prototype-based learning, which is a well-established technique (e.g., in metric learning). Could the authors clarify what the core contribution of the paper is beyond these existing ideas?
>
> A2: Although attention masking and prototype optimization have been explored in other studies, our proposed GPR-NIAM framework is the first to integrate Non-Interfering Attention Masking (AttnIso) with Cross-Silo Collaborative Refinement (CSCR) in a unified manner, specifically tailored for the one-shot Federated Prompt Learning (FPL) setting. (1) Unlike conventional prompt-tuning methods where learnable prompts and text tokens interact freely, **the AttnIso module employs a non-interfering attention masking strategy that enforces unidirectional information flow through hard masking and λ-weighting.** This design not only preserves transferable knowledge but also enhances cross-task generalization, which is particularly crucial in the one-shot FPL setting. (2) Although prototype learning has been explored in prior studies (e.g., FedProto, FedProc), **these methods typically rely on aggregating local prototypes from clients to form a global prototype, which requires multiple rounds of interaction and thus cannot be applied in one-shot scenarios.** In contrast, **our approach fully leverages the potential of pre-trained models by freezing the visual encoder, which effectively reduces feature distribution discrepancies across clients.** This design enables our CSCR module to directly perform centralized optimization and alignment of the global prompt using multi-source visual prototypes on the server side.
>
> W3: The paper claims that previous methods neglect the preservation of generalization to unseen tasks. However, it is not clear why the proposed text-prompt masking mechanism and the visual representation-guided prompt are effective in addressing this issue. Could the authors clarify the connection between these two modules and explain how they jointly contribute to improving generalization?
>
> A3: Our design aims to balance task-specific adaptation with the retention of transferable knowledge to enhance generalization to unseen tasks. **The AttnIso module employs non-interfering attention masking to enforce unidirectional information flow, preventing learnable prompts from excessively interfering with the original text embeddings and thus preserving the general semantic structure of the pre-trained model.** Meanwhile, **the CSCR module leverages multi-source visual prototypes to perform centralized optimization and alignment of the global prompt on the server side, effectively mitigating the bias caused by client data heterogeneity.** The combination of these two modules enables the model to avoid overfitting to seen tasks at the local level while enhancing cross-domain representation capabilities globally, leading to stronger generalization on unseen tasks as verified by our experimental results.
>
> W4: In Figure 1, the meanings of T_T, T_P, and T_EOS are unclear. Why can the attention maps of the three mechanisms be shown in a single figure? What exactly are the differences between the three masking strategies, and what results or benefits does each masking mechanism correspond to?
>
> A4: $T_T$ denotes the original text tokens (e.g., ``a photo of a [CLASS]''), $T_P$ represents the learnable prompt tokens, and $T_{\mathrm{EOS}}$ is the end-of-sequence token used to aggregate contextual information. Figure 1 shows the three mechanisms together because they are all controlled by a unified attention mask matrix $M$, corresponding to full attention ($0$), hard masking ($-\infty$), and $\lambda$-reweighting ($\lambda$), respectively. Full attention allows unrestricted token interaction but may distort the original semantics; hard masking blocks the influence of $T_P$ on $T_T$, preserving the pre-trained semantic structure; and $\lambda$-reweighting enables controlled information flow between $T_{\mathrm{EOS}}$ and $T_T$, balancing task adaptation with knowledge retention. The combination of these strategies avoids overfitting while enhancing cross-task generalization, as verified in the ablation results (Table~3).
>
> W5: I noticed that the baseline results reported in the experimental section are inconsistent with those in the original papers. For example, in Table 5, the reported performance of FedTPG differs from that in the original work across several datasets. Could the authors clarify the reasons behind these discrepancies?
>
> A5: The discrepancies mainly arise because we re-implemented all baseline methods under a unified Federated Prompt Learning (FPL) framework to ensure fair comparisons. For example, the original FedTPG paper was not evaluated in a one-shot FPL setting, so we adapted it to a single-round communication protocol, which inevitably affects the results. In addition, we used a consistent setup with the pre-trained CLIP (ViT-B/16) backbone and a fixed prompt length (10 × 512), and simulated non-IID data across multiple clients using a Dirichlet distribution ($\beta = 0.5 or 0.1$), which differs from the configurations in the original papers. Furthermore, some baseline implementations were unavailable or incompatible with our experimental environment, and we re-implemented them based on the original descriptions, which may lead to minor deviations.

---

> > ### Comment · Reviewer_bm2r · 2025-08-04
> > **Reply**
> >
> > Thank the authors for answering my concerns very clearly, and I also read the positive comments from other reviewers.
> > Therefore, I would like to increase my score.

---

> > > ### Author Response · Authors · 2025-08-06
> > > **Thanks for your feddback**
> > >
> > > Thank you for your valuable feedback. Your review provides important guidance and inspiration for our future work.

---

### Comment · Area_Chair_uP5U · 2025-08-02
**Friendly Reminder: Engaging with Author Rebuttals**

Dear Reviewer,

Thank you for your time and expertise in reviewing for NeurIPS 2025. As we enter the rebuttal phase, we kindly encourage you to:

1) Read and respond to authors' rebuttals at your earliest convenience.

2) Engage constructively with authors by addressing their clarifications in the discussion thread.

3) Update your review with a "Final Justification" reflecting your considered stance post-rebuttal.

Your active participation ensures a fair and collaborative evaluation process. Please don’t hesitate to reach out if you have any questions.

With gratitude,

Your AC

---

### Decision · Program_Chairs · 2025-09-17

**Decision:**

Accept (poster)

**Comment:**

This paper proposes GPR-NIAM, a method for one-shot federated prompt learning that uses a novel attention masking mechanism and cross-silo collaborative refinement to improve communication efficiency and cross-task generalization without multi-round communication. The four reviewers' ratings are: Weak Accept, Weak Accept, Accept, Accept. The strong consensus on its empirical rigor and the significance of enabling one-shot federated learning for prompt-based methods aligns with NeurIPS's standards.